



# A novel model for simulation of nitrate in aquifers

Roohollah Noori[1], Mehrnaz Dodangeh[1], Ronny Berndtsson[2], Farhad Hooshyaripor[3], Jan Franklin Adamowski[4], Saman Javadi[5], Akbar Baghvand[1]

[1]Department of Environmental Engineering, Graduate Faculty of Environment, University of Tehran, Tehran, 1417853111, Iran
[2]Department of Water Resources Engineering & Center for Middle Eastern Studies, Lund University, Box 118, SE-221 00 Lund, Sweden
[3]Department of Civil Engineering, Science and Research Branch, Islamic Azad University, Tehran, 1477893855, Iran
[4]Department of Bioresource Engineering, Faculty of Agricultural and Environmental Sciences, McGill University, Montreal, H9X 3V9, Canada
[5]Department of Irrigation and Drainage, Aburaihan Campus, University of Tehran, Tehran, 3391653755, Iran

*Correspondence to*: Roohollah Noori (noor@ut.ac.ir)

**Abstract.** Please Numerical groundwater quality models (GQMs) often run at high computational cost resulting in long simulation times and complex parameter calibration that limit their practical applications. In this study, a novel reduced-order model (ROM) was developed for nitrate simulation in groundwater including a simple structure and with similar accuracy as more extensive GQMs. The proposed methodology for the development of ROM presents a solution for the problem in ROMs developed with eigenvectors, to make predictions into the future. The model performance was investigated by simulation of nitrate in the Karaj Aquifer, Iran. The dominant modes of spatiotemporal variation of nitrate during a five-year period was calculated by the model. The results revealed an excellent agreement between nitrate simulated by the ROM and the well-known Modular Transport 3D Multi Species (MT3DMS). The absolute error between the ROM and the MT3DMS was less than 0.5 mg/l in the most parts of the aquifer. Thus, results confirm that the use of ROM has advantages through a much simpler structure and shorter calculation times. Observed spatiotemporal variation of nitrate in the aquifer was well represented by the ROM simulations. The simplicity of the model makes it highly interesting also to other water resources problems.

## 1 Introduction

High dependency on groundwater resources and excessive water withdrawal from aquifers have led to drastic drop in groundwater levels in many parts of the world. In addition, agricultural, industrial, and urban activities may be resulted in pollution spread in the aquifers that finally result in groundwater quality deterioration. In this regard, nitrate is often a main concern for most aquifer water quality. Studying the spatiotemporal variation (STV) of nitrate in aquifers provides important information for effective management of the contaminated groundwater resources and possibilities for future improvement. Due to lack of proper groundwater monitoring systems as well as small research budgets especially in developing countries, groundwater quality simulation models (GQSMs) are essential for evaluation of the STV of nitrate in these aquifers. The



models can provide important information regarding the present status and implications of future scenarios regarding the STV of nitrate contamination.

A literature review clearly demonstrates the successful applications of one of the most popular GQSMs, the Modular Transport 3D Multi Species (MT3DMS) (Zheng and Wang, 1999; Conan et al., 2003; Peña-Haro et al., 2009 and 2010; Saghravani et

al., 2011; Rasmussen et al., 2013; Gusyev et al., 2014; Abdelaziz and Merkel, 2015; Pulido-Velazquez et al., 2015; Laattoe et al., 2017). Three important challenges, however, as discussed below, influence such models regarding management decisions based on their results: (A) In many cases (especially the use of three-dimensional models with long simulation times for pollutants in extended aquifers), the computational costs are very high and the large amount of information produced by the models is confusing (Li et al., 2004; Vermeulen et al., 2005; Stanko et al., 2016). This poses challenges regarding decision

making and proper management of the aquifers. In such cases, application of alternative methods such as reduced-order models (ROMs) based on proper orthogonal decomposition (POD) may be an alternative (Esfahanian and Ashrafi, 2009). The accuracy of ROMs has been shown to be compatible with GQSMs while ROMs pose a much simpler structure (Cardoso et al., 2009; Ushijima and Yeh, 2017). In ROMs based on eigenvectors the dominant spatiotemporal modes of the target are calculated. Using the few first dominant modes, a simple model is developed to simulate the target. Successful application of this approach

in the field of water resources management (beyond the laboratory scale in which many simplifying assumptions may be employed) has been reported for simulation of nitrate and water temperature in the Karkheh Dam Reservoir located in Iran (Noori et al., 2015 and 2018) and surface currents in the Gorgan Bay, Iran (Kheirabadi et al., 2018). Some other studies have aimed at applying ROMs in subsurface flow (Vermeulen et al., 2004, 2005 and 2006; McPhee and Yeh, 2008; Siade et al., 2010; Pasetto et al., 2011, 2013 and 2014; Stanko et al., 2016), however, no applications have been reported using them to

quantify pollutants in aquifers. Due to the difficulty of ROMs to make predictions into the future as described by Noori et al. (2017), the above work (Noori et al., 2015) only aimed at regeneration of nitrate in the reservoir during the simulation period. Therefore, the most important goal of this study was to develop a ROM based eigenvectors that can predict the future concentration of nitrate in the aquifer (beyond the observation period). (B) The second challenge in application of GQSMs is their limitation in providing the dominant modes of the pollutant variation in the simulation period. GQSMs can provide STV

of pollutants in the aquifer and deliver useful information for water quality managers. But, when the aim is determining the dominant modes of the pollutant, the user needs to use an alternative solution like POD linked to the GQSMs (Kostas et al., 2005; Bennacer and Sefiane, 2016). The POD can use snapshots of pollutant distribution simulated by GQSMs, thus, determining the dominant modes of pollutant variation. These dominant modes appropriately present the most representative patterns of STV of pollutant in the simulation period (Kang et al., 2015). To the best of the authors' knowledge, no application

has yet been reported aiming to present dominant modes of STV for pollutants in groundwater systems. (C) Finally, the third challenge for application of QGSMs is the complex mathematical form of the governing equations as well as their complicated solving methods (Mehl, 2006; Stanko et al., 2016; Pasetto et al., 2017). Fortunately, the ROMs overcome this problem to a large extent by providing a model with much simpler mathematical form than the QGSMs whilst they keep an acceptable level of accuracy (Boyce et al., 2015).



In view of the above, in the present study, we modify the developed nitrate ROM with eigenvectors so that it enables prediction of nitrate STV in the Karaj Aquifer in the future time. The dominant modes of the nitrate variation in the aquifer are calculated and presented. In this regard, the groundwater flow in the study area was firstly simulated by MODFLOW (Harbaugh et al., 2000). Thereafter, STV of nitrate was calculated by MT3DMS model. Finally, the nitrate ROM was developed for the aquifer.

The presented methodology in this study is novel and it has broad applications in other fields of science and engineering.

## 2 Methods

### 2.1 Study area

The study area is the large unconfined groundwater Karaj aquifer with an area of 175.6 km$^2$ located in the southern parts the Alborz Mountain chain (Figure 1). The mean annual rainfall from 1967 to 2012 was 285.5 mm, with a coefficient of variation

of 30.3%. The minimum and maximum annual values of rainfall were 93.3 and 452.6 mm, respectively, whilst the average annual evaporation was about 2430 mm. The only permanent river in the study area is the Karaj River with an average annual flow of 470 million cubic meters (MCM) that flows through the eastern region of the aquifer. The northern part is mountainous including high-altitude areas that are generally covered with fine-grain alluvial deposits, while the low-altitude areas in the southern part are coarse-grained alluvial deposits. In general, the study area has a relatively sharp slope with north and northeast

to south and southwest direction. It results in variation of land surface topography in the study area from 1381 m in the north-east to 1200 m in the south-west of the plain.

Due to its proximity to the Tehran metropolis, Karaj city has a high population density that has led to increasing water demand. To meet the water demand, groundwater withdrawal has increased resulting in a significant drop in water level and also groundwater quality degradation especially in the recent years. There are 12 observation wells (piezometers) in the study area

(Figure S1). The water level monitoring in these wells, clearly shows that the aquifer is experiencing a decreasing trend (typically 1.14 m per year water level decrease from April 2006 to April 2012).

### 2.2 Groundwater flow and nitrate transport models

To simulate the groundwater flow and nitrate transport in the Karaj Aquifer, MODFLOW and the Modular Transport 3D Multi Species (MT3DMS) models (Harbaugh et al., 2000; Zheng and Wang, 1999), respectively, were used. Figure 2 shows the

25 different steps of conducting the present study. MODFLOW is a three-dimensional (3D) model originally developed by the US Geological Survey that uses a block-centered finite difference technique to solve the groundwater flow equations in the saturated aquifers. The general governing equation solved by MODFLOW is:

$$\frac{\partial}{x_i}\left(K_i \frac{\partial h}{x_i}\right) + q_s = S_s \frac{\partial h}{\partial t} \tag{1}$$





where $K_i$ is hydraulic conductivity along $i$ axis (m/s), $x_i$ is the distance along the respective Cartesian coordinate axis (m), $q_s$ is volumetric flux per unit volume (1/s), $S_s$ is specific storage (1/m), $h$ is the potentiometric head (m), and $t$ is time (s) (Abdelaziz and Zambrano-Bigiarini, 2014; Abdelaziz and Le, 2014).

The flow pattern simulated by MODFLOW was used as input to MT3DMS, developed by Zheng and Wang (1999), to simulate
the nitrate concentration in the aquifer. The partial differential equation describing the transport of contaminants in 3D transient groundwater flow systems can be written as:

$$\frac{\partial (nC)}{\partial t} = \frac{\partial}{\partial x_i}\left(nD_{ij}\frac{\partial C}{\partial x_i}\right) - \frac{\partial}{\partial x_i}(nv_iC) + q_sC_s + \sum R_n \qquad (2)$$

where $C$ is dissolved concentration of nitrate (g/l), $n$ is porosity of the medium, $D_{ij}$ is hydrodynamic dispersion coefficient tensor (m²/s), $v_i$ is the seepage or linear pore water velocity (m/s) calculated according to Eq. (3), $C_s$ is concentration of the
source or sink flux (g/l) and $\sum R_n$ is chemical reaction term (g\l-s).

$$v_i = -\frac{K_i}{n}\frac{\partial h}{\partial x_i} \qquad (3)$$

MODFLOW requires aquifer geometry, initial estimate of hydrodynamic parameters (hydraulic conductivity and storage coefficients), discharge of operational wells and their returned flow, amount of recharge, piezometers' information, and boundary and initial conditions as inputs. MT3DMS, in addition to the MODFLOW outputs, needs dispersion coefficients and
nitrate loads as initial and boundary condition.

According to geological studies, the Karaj Aquifer is a single-layer unconfined aquifer. The aquifer's thickness in different locations is illustrated in Figure S2. It varies from 86 to 347 m as presented in the figure. Evaluation of the water table in the piezometers reveals that the groundwater flows from north and northwest to the south and southwest of the aquifer. Thus, the boundary conditions are defined as two boundary lines including given heads at the north-western part of the aquifer for the
entrance and at the southern part of the aquifer for the outflow (red lines in Figure S1). For instance, the iso-water table for April 2011 is shown in Figure S1. For the other boundaries there is no interaction between the adjacent aquifers.

The spatial distribution hydraulic conductivity was obtained by the Ministry of Energy that estimated these parameters from pumping tests. Initial values of specific yield were chosen on the basis of the guideline suggested by Johnson (2012). The final values of the hydraulic conductivity and specific yield are determined during the calibration stage.

There are 1,698 production wells used for different purposes such as industrial and agricultural activities (Figure S1). Using the information from these wells and land use (Figure S1), the groundwater discharge and water return coefficients for each land use type were calculated. The evaporation from groundwater was considered negligible since the groundwater table is deeper than three meter in all parts of the aquifer.

Besides natural rainfall infiltration, seepage from pit latrines is one of the sources of aquifer recharge due to vast use of this
system for wastewater disposal in the residential areas. The recharge amount from pit latrines was estimated based on population density and land use maps included in the calibration procedure. Karaj River is the other source of groundwater



recharge in the study area. Note that at the entrance point of the river into the plain, part of water is diverted and transferred by a channel to supply drinking water of the surrounding towns. According to available data, mean annual depth of the river at the entry point to the plain for 2006-2008, 2008-2010, and 2010-2012 was 76.7, 92.8, and 66.7 cm, respectively. At the outlet of the river from the plain, the river had typically insignificant flow for all years.

The spatial distribution of nitrate in April 2006 was used as initial conditions for the transport model. By using estimates of water recharge from different agricultural, industrial, and residential land uses and nitrate concentrations in recharged water, the nitrate loads into the aquifer were calculated. The nitrate loads were then calibrated according to those measured in the piezometers.

## 2.3 Calibration and verification

A six-year period from April 2006 to March 2012 was selected for calibration of the flow (MODFLOW) and nitrate transport (MT3DMS) models under unsteady state condition. This period was selected due to including the most accurate and complete data such as meteorological, hydrological, geological, hydrogeological, and water quality data. In addition, April 2011 was considered for calibration of the groundwater flow model under steady state condition. To verify the unsteady models' robustness, monthly data from April 2012 to March 2013 was employed.

### 2.3.1 Groundwater flow model (MODFLOW)

The aquifer was divided into a 100×100 gridded network with 217.9×183 m mesh dimension. The number of active and inactive cells in the model were 4,691 and 5,309, respectively. In order to calibrate the model under steady state conditions, firstly, the piezometer data in April 2011 were used in the model as initial values. Note that spatially distributed groundwater levels for the aquifer were generated from piezometer data by kriging. In the next step, the model's calibration was performed
manually by adjustment of hydraulic conductivities. Input recharge was affected by great uncertainty, thus, they were simultaneously modified in the calibration process. The calibration process was continued until the Mandle (2002) criterion was met (difference between simulated and observed hydraulic heads less than 10% of the domain variation). The maximum domain variation was considered 10 m. Spatial variation of calibrated hydraulic conductivity and recharge are shown in Figures S3 and S4, respectively. According to the figure, highest hydraulic conductivity is more than 45 m/day in the southwest part,
while minimum is about 4 m/day in the northwest. Spatial distribution of the calibrated recharge indicates that the urban densely populated areas experience high recharge values of more than 10 mm/day. The widespread using of pit latrines and wells for wastewater disposal may contribute the high rate of recharge in the urban areas.

In the next step the specific yields were calibrated under unsteady state condition. In this process, monthly time series of calibrated recharge, piezometer, and operational well data were used as model input. Considering 10 m accuracy of hydraulic
head in the model domain, the spatial variation of calibrated specific yields is shown in Figure S5. According to the figure, the specific yield varies from 0.15 in the south to 0.01 in the north of the aquifer.

After calibration, the model was verified for the period April 2012 to March 2013.



## 2.3.2 Contaminant transport model (MT3DMS)

In MT3DMS model, there are five methods for solving the governing partial differential equations: method of characteristics, modified method of characteristics, hybrid method, finite difference, and ultimate methods. In the present study, the ultimate method, which is a combination of four other methods, was used to discretize the advection-diffusion equation. The model

was calibrated with trial and error. For this purpose, nitrate concentrations in April 2006 were used as initial conditions. Then, calibration of the model during the six-year period was done by modifying dispersion coefficients and nitrate input loads. Note that as the aquifer is single layer and unconfined, only longitudinal and transverse dispersion coefficients were calibrated in the model. Due to the fact that porosity is almost equal to the specific yield in unconfined course-graded aquifers, the calibrated specific yield was used for porosity in the calibration of the MT3DMS model. A threshold of 10% of nitrate variation during

the simulation period was attained (here 10 mg/l).

After calibration of the MT3DMS, verification was carried out for a one-year period (April 2012 to March 2013).

## 2.4 Nitrate ROM

After calibration and verification of the flow and nitrate transport models for the Karaj Aquifer, the nitrate ROM was developed. A POD model was used in a first step to determine dominant modes of nitrate variation for the Karaj Aquifer. The

POD was applied to nitrate concentration simulations by MT3DMS for the aquifer ($\mathbf{\Omega}$) resulting in spatial components of dominant modes of nitrate concentrations ($\mathbf{\Theta}$) given by:

$$\mathbf{\Theta} = \sum_{i=1}^{N} \xi \mathbf{\Omega}^{(i)}(\mathbf{x}) \tag{4}$$

In Eq. (4), $N$ is the number of snapshots, $\mathbf{x}$ is the location matrix of cells, and $\xi$ is the eigenvectors (Ashrafi, 2012). This relationship is a linear combination of snapshots taken from nitrate simulated in the Karaj Aquifer. To increase the similarity

between functions $\mathbf{\Theta}$ and $\mathbf{\Omega}$, it is necessary to determine $\xi$ in such a way that the following expression is optimized (Noori et al., 2015):

$$\frac{1}{N} \sum_{i=1}^{N} \left\{ \left| \left( \mathbf{\Omega}^{(i)}, \mathbf{\Theta} \right) \right|^{2} / (\mathbf{\Theta}, \mathbf{\Theta}) \right\} \quad if \quad (\mathbf{\Theta}, \mathbf{\Theta}) = \|\mathbf{\Theta}\|^{2} = 1 \tag{5}$$

The optimization of Eq. (5) is done by solving the following eigenvalue problem:

$$\sum_{j=1}^{N} \mathbf{M}_{ij} \xi_{j} = \lambda \xi_{i} \tag{6}$$

where $\mathbf{M}$, $\lambda$, and $\xi$ are Hermitian matric, eigenvalues, and their corresponding eigenvectors, respectively (Noori et al., 2015). Having the function $\mathbf{\Theta}$ and using Eq. (7), the temporal components of dominant modes of nitrate in the Karaj Aquifer can be calculated (Ashrafi, 2012):

$$\tau_{i}(t) = (\mathbf{\Omega}(\mathbf{x}, t), \mathbf{\Theta}_{i}(\mathbf{x})) \tag{7}$$



It is noteworthy that all eigenvalues calculated by Eq. (6) are positive. Only the few first calculated eigenvalues are large and the rest are close to zero. Since the calculated modes are corresponding to eigenvalues, only the few first modes are important in practical problems so that they represent most of the system variation (STV of nitrate concentration in the aquifer). Thus, it is possible to generate the STV of nitrate concentration in the aquifer by means of the first modes and application of the

following equation (Noori et al., 2017):

$$\mathbf{\Omega}(\mathbf{x}, t) \cong \sum_i^l \tau_i(t) \mathbf{\Theta}_i(\mathbf{x}) \quad , \quad l \ll N \tag{8}$$

where $l$ is the number of modes.

Eq. (8) has a simple structure with a few modes. Note that to be able to develop the nitrate concentration ROM by application of Eq. (8), one need simulated nitrate concentrations from the MT3DMS. In order to circumvent this problem, we develop a

methodology that enables the ROM to independently simulate nitrate concentrations. For this purpose, it is necessary to calculate both the spatiotemporal components $\mathbf{\Theta}(\mathbf{x})$ and $\tau(t)$ for the future time interval $(t + n)$. Since the component $\mathbf{\Theta}$ is a function of space, it does not change in time. Therefore, it is necessary to calculate $\tau$ for future time steps. For this purpose, a regression equation was used to estimate the time variation of this component ($\tau(t+n)$). Having $\tau(t+n)$, the nitrate concentration for future time steps $t+n$ can be calculated by:

$$\mathbf{\Omega}(\mathbf{x}, t + n) \cong \sum_{i=1}^l \tau_i(t + n) \mathbf{\Theta}_i(\mathbf{x}) \quad , \quad l \ll N \tag{9}$$

Eq. (9) simulates the nitrate concentration in different parts of the aquifer for future time t+$n$ using the first few modes.

## 3 Results and discussion

### 3.1 Results of groundwater flow model

The calibration of MODFLOW was continued by six-year data from April 2006 to March 2012 until the difference between

simulated and observed groundwater levels for steady state condition in all piezometers was less than 10% of the hydraulic load variation within the domain (i.e., 10 m). In addition, the difference between simulated and measured groundwater levels during the groundwater calibration for unsteady conditions are shown in Figures 3A and 3B for two piezometers. As seen from the figures, simulated piezometer levels match the observations well. Errors during calibration period were less than 10% of the hydraulic head variation (i.e., ±10 m) as suggested by Mandle (2002).

After successful calibration of the groundwater flow model, its performance was verified using data from April 2012 to March 2013 as shown in Figure 3C. According to the figure, the temporal variation of mean absolute error (MAE) and root mean square error (RMSE) is between 2.6 and 4.2 m. Therefore, it can be concluded that the groundwater flow model was reliably calibrated and verified for the aquifer.





## 3.2 Results of nitrate transport model

To properly tune the model, a threshold value of 10% of nitrate variation in the simulation period would be acceptable (here 10 mg/l). Figures 3D to 3F show the difference between simulated vs measured nitrate concentration during the calibration of MT3DMS under unsteady conditions in three selected piezometers for the Karaj Aquifer. According to these figures,

MT3DMS calibration has been performed well so that the error between simulated vs measured nitrates is less than 10 mg/l. Verification of the MT3DMS model was carried out using one-year data from April 2012 to March 2013. In this step, mean values of MAE and RMSE were equal to 2.3 and 2.9 mg/l for nitrate concentration, respectively. The spatial distribution of nitrate concentration in the Karaj Aquifer for six selected months is illustrated in Figures 4A to 4F. The figures show that the northern part of the aquifer are more exposed to the risk of nitrate contamination. Nitrate concentration in some parts exceeds

the maximum permissible limit 50 mg/l suggested by World Health Organization. A look at the land use map (Figure S1) indicates that these parts include densely populated urban areas that typically use pit latrines for wastewater disposal.

## 3.3 Dominant modes of nitrate STV

To calculate the dominant modes of nitrate STV in the Karaj Aquifer, snapshots of nitrate concentrations simulated by MT3DMS were used from the calibration period. Due to the low variation of nitrate in groundwater compared to surface water,

one-day intervals were selected for capturing variation. Consequently, according to the six-year simulation period, 1,772 snapshots of nitrate concentration were obtained for 4,691 active cells of the Karaj Aquifer. In other words, a matrix $\Omega$ was formed with the dimension 1,772×4,691 as described in "Methods" section. By applying the POD on the matrix $\Omega$, firstly the Hermitian matrix was formed with dimension equal the number of snapshots, i.e., 1,772×1772. Thereafter, the eigenvalues and their corresponding eigenvectors were calculated. The results for the first ten eigenvalues are shown in Figure 5A. The

energy of the system conserved (STV of nitrate concentration in the aquifer) for the first ten eigenvalues is shown in Figure 5A. The results show that except the first eigenvalue that conserves about 96.2% of the energy of the system, the others are negligible. Thus, only the first mode may represent the STV of nitrate concentration in the Karaj Aquifer. Temporal and spatial components of the first mode ($\tau_1(t)$ and $\Theta_1(x)$) are shown in Figures 5B and 5C, respectively. According to Figure 5C, it is clear that the largest variation of nitrate occurs in the northern parts of the aquifer (shown in orange and blue color) in

connection to urban areas as mentioned above. Indeed, high nitrate loads are introduced into the groundwater from residential areas due to the lack of sewage collection systems and using pit latrines for wastewater disposal. According to Figure 5B, the first temporal component ($\tau_1(t)$) indicates a gradually increasing trend for nitrate concentration in the aquifer. This is due to the increase in population and development of agricultural lands, especially in recent years, and consequently increasing nitrate loads into the Karaj Aquifer. Figure 5B also indicates that the temporal variation of the other components is relatively

insignificant.



### 3.4 Results of nitrate ROM

### 3.4.1 Developing ROM for nitrate calculation

In the study, two important principles were considered for the development of the nitrate ROM: (A) The accuracy should be similar to the results of MT3DMS, and (B) the model structure should be kept simple. For this purpose, a trade-off between
the model´s accuracy and its complexity was performed. Figure 5D shows the RMSE for simulated nitrate concentrations by the MT3DMS and different ROMs developed with different number of modes. According to the figure, it is clear that increase in number of modes is accompanied by an increase in accuracy of the ROM. This increase in the models' accuracy is meaningful until the ROM developed by the first ten modes. Therefore, the nitrate ROM developed by the first ten out of 1,772 modes were used for simulation of nitrate in the aquifer. These ten modes together represent more than 99.99999% of the STV
of nitrate in the aquifer. The developed nitrate ROM was applied to simulate the nitrate concentrations for the aquifer during the calibration period. The spatial distribution of the simulated nitrate concentration by ROM and MT3DMS, as well as the difference between results of the two models for four selected days is presented in Figures 6A to 6D. According to the figures, the difference between results of the two models in different parts of the aquifer is less than 0.1 mg/l nitrate. This value is very small as compared to the large observed variation in nitrate concentration in the Karaj Aquifer (maximum nitrate concentration
is about 116 mg/l). The mean error for each cell during the other four selected days are shown in Figures 6E to 6H. The graphs show that the error varies between of -0.4 and 0.4 mg/l nitrate concentration. To better evaluate the accuracy of the ROM, nitrate concentrations simulated by ROM vs MT3DMS for the four different days are shown in Figures 6I to 6L. According to the figures, both models perform similar giving a correlation of about 1 for each of the days. Thus, it can be concluded that both models are equal in terms of accuracy.

### 3.4.2 Developing ROM for prediction purposes

In order to develop ROM for prediction of nitrate concentration beyond the simulation period, it is necessary to calculate the temporal components of the first ten modes for future time step $t+n$. In this regard, one-year data from April 2012 to March 2013 was used. However, given the high importance of the temporal component of the first four modes that represent more than 99.99% of STV of nitrate in the aquifer, only the trend of these components are estimated in the future times $t+n$.
According to Figure 5B, for each component the best models were fitted. The fitted models with $R^2$ equal to 0.999, 0.995, 0.959, and 0.892 for the first four temporal components, respectively, can be used to properly reconstruct these components for future time steps with good accuracy. The other temporal components were ignored due to their small contribution. Thereafter, having the spatial components of the first four modes, ROM was developed by application of Eq. (9) in the "Methods" section. Figures 7A to 7C show the spatial distribution of nitrate concentration simulated by ROM and MT3DMS,
respectively, as well as the difference between the model results for three selected days in 2012. According to the figures, maximum spatial difference between the models is less than 3 mg/l nitrate concentration. The results indicate that the absolute error is less than 0.5 mg/l nitrate concentration in the most parts of the aquifer. Thus, both models behave in a similar manner.

It is noteworthy that the accuracy of the prediction is slightly lower than that obtained by ROM developed by the first ten modes. This is due to the use of the only first four out of first ten modes for development of ROM that conserved a lower percentage of the system energy compared to the first ten modes.

## 4 Conclusions

Some limitations of GQSMs were the main motivation for us to explore and present an alternative method to overcome these problems. In this regard, a ROM was developed by linking between MT3DMS and POD models and its performance was checked by simulation of nitrate concentration in the Karaj Aquifer, Iran. The developed nitrate ROM provided desirable results that matched the nitrate simulated by MT3DMS model well. More specifically, dominant variation of nitrate, based on the results obtained by spatial distribution of the first mode of nitrate, revealed the largest variations of nitrate in the northern parts of the aquifer where occupied by dense population that using the septic wells for wastewater disposal. Also, the temporal component of the first mode indicated a gradually increasing trend of nitrate during the simulation period in the aquifer. The absolute error between the ROM and the MT3DMS was very small in most parts of the aquifer (about 0.5 mg/l nitrate concentration) compared to the large observed variation in nitrate concentration (maximum nitrate concentration was about 116 mg/l). In fact, the developed ROM with a much simpler structure had the same performance compared with MT3DMS. Therefore, it can be concluded that the developed ROM was superior than MT3DMS. Indeed, the introduced methodology is general and it can be applied for simulation of target parameters in other fields of study.

**Author contribution**

Roohollah Noori and Mehrnaz Dodangeh conceived the study, collated the data, and ran the model. Farhad Hooshyaripor, Jan Franklin Adamowski, Saman Javadi, and Akbar Baghvand supervised the result analyses. Also, all authors contributed to the writing and reviewing of the manuscript.

**Supporting information**

SI includes: Figure S1: observation wells, land use map, production wells, and groundwater table in the aquifer, Figure S2: thickness of the aquifer, Figure S3: spatial variation of calibrated hydraulic conductivity in the aquifer, Figure S4: spatial variation of recharge in the aquifer, and Figure S5: spatial variation of calibrated specific yields in the aquifer.

**Conflict of interest**

The authors declare no conflict of interest.





**Acknowledgments**

This research was funded by the Center for Middle Eastern Studies, Lund University. The authors also are gratefully acknowledging the Iran Water Resources Management Company for cooperation in data preparation.

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





**Figure Captions**

**Figure 1:** The Karaj Aquifer located in northern Iran

**Figure 2:** Methodology used for the present study

**Figure 3:** Difference between simulated and observed groundwater levels during the calibration of MODFLOW: (**A**) under unsteady state condition for Ghezel-hesar piezometer located in the northwestern part of the aquifer, (**B**) under unsteady state condition for Marlik piezometer located in the southern part of the aquifer; (**C**) Temporal variation of mean absolute error (MAE) and root mean square error (RMSE) during the one-year verification of MODFLOW; Difference between simulated and observed nitrate concentration during the calibration of MT3DMS under unsteady state condition in: (**D**) Golshahr piezometer, (**E**) Mehrshahr piezometer, and (**F**) Meshkin piezometer

**Figure 4:** Spatial distribution of nitrate concentration in the Karaj Aquifer for six selected days (**A**) 15th September 2006, (**B**) 15th April 2007, (**C**) 15th July 2008, (**D**) 15th November 2009, (**E**) 15th August 2010, (**F**) 15th February 2011

**Figure 5:** (**A**) Eigenvalues and corresponding eigenvectors with energy of the system conserved for the first ten eigenvalues calculated by application of POD, (**B**) temporal components of first to fourth modes {$\tau_1(t)$ to $\tau_1(t)$}, (**C**) spatial component of the first mode {$\Theta_1(x)$}, (**D**) root mean square error (RMSE) for nitrate concentrations simulated by MT3DMS and ROMs with different number of modes

**Figure 6:** Spatial distribution of nitrate concentrations simulated by ROM, MT3DMS, and difference between the two model results for four selected days: (**A**) 15th September 2006, (**B**) 10th June 2007, (**C**) 10th April 2008, (**D**) 10th January 2010; Mean error in each cell for four selected days: (**E**) 15th November 2006, (**F**) 12th May 2008, (**G**) 14th September 2009, (**H**) 08th February 2010; Nitrate concentrations simulated by ROM vs MT3DMS for four different days: (**I**) 05th December 2006, (**J**) 26th August 2007, (**K**) 20th April 2009, (**L**) 12th May 2010

**Figure 7:** Spatial distribution of nitrate concentrations simulated by ROM and MT3DMS, and difference between the models for three selected days beyond the simulation period: (**A**) 21th May 2012, (**B**) 18th September 2012, (**C**) 04th February 2013





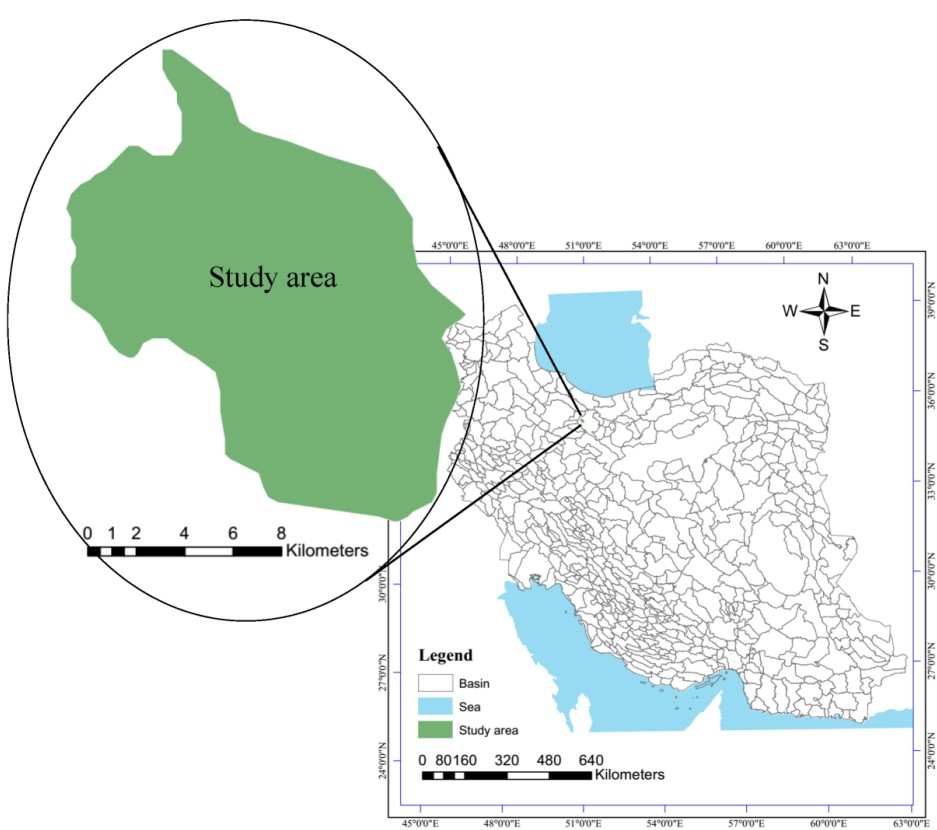

**Figure 1**







**START**

**Model Construction**

**Development of nitrate ROM**

Nitrate ROM

Formation of eigenvalue problem

Establishment of Hermitian matrix

Calculation of the eigenvalues and eigenvectors

Calculation of temporal and spatial terms of ROM {a(t) and Φ(x), respectively}

Selection of the number of modes that properly describe nitrate variation in the system

Developing the ROM for simulation of nitrate in the system

Regeneration of nitrate using developed ROM in the system

**Development of groundwater flow and nitrate transport models**

First: Groundwater modeling

Data preparation

Model selection (MODFLOW and MT3DMS)

Preparing the conceptual and mathematical models

Create a network grid model

Converting the conceptual model to the network model

Groundwater flow model (MODFLOW)

Transient flow modeling for 6 years

Calibration for 6 years

Verification for a year

Steady-state flow modeling for a year

Calibration for a year

Running MT3DMS

Nitrate calibration for 6 years

Determination of the spatiotemporal intervals to extract the simulated results by MT3DMS

Coding in MATLAB environment to convert the outputs of MT3DMS into a usable format for POD application

Extraction of MT3DMS results

Comparing between simulated and regenerated nitrate by MT3DMS and ROM, respectively

Yes ← Satisfying the defined criteria? → No

**Model Verification**

Development of nitrate ROM to simulate unseen nitrate data

Calculation of temporal component of nitrate ROM in future time by a regression method

Nitrate simulation in the aquifer using developed ROM in future time

Comparing between simulated nitrate by MT3DMS and ROM

Yes ← Satisfying the defined criteria? → No

**END**

**Figure 2**





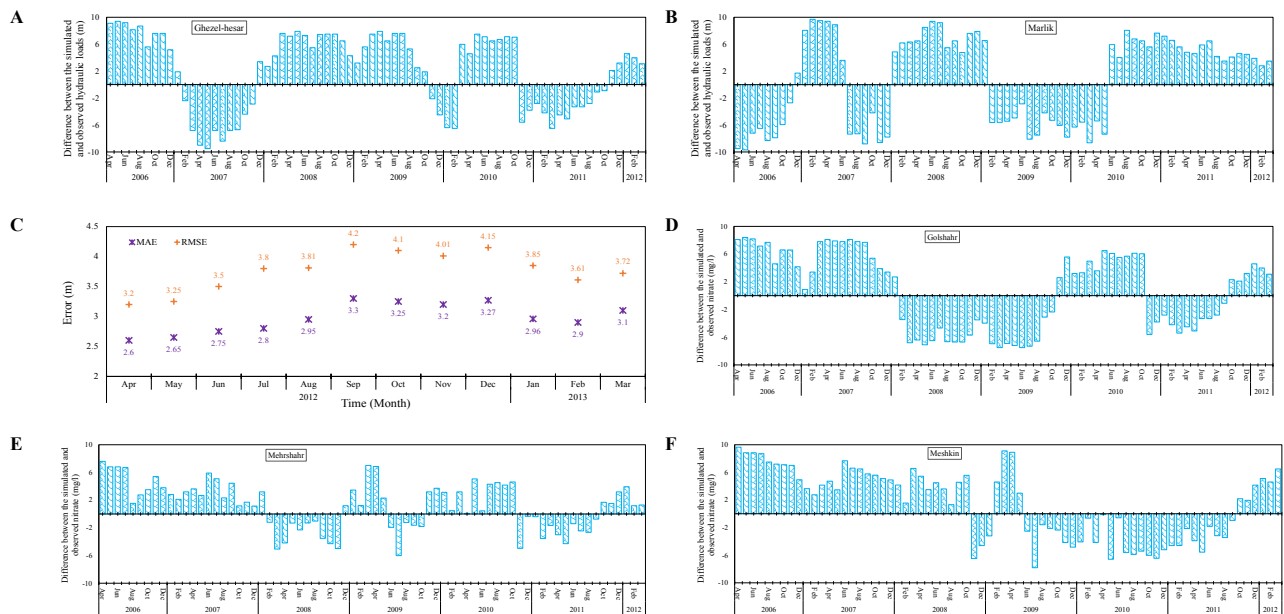

**Figure 3**



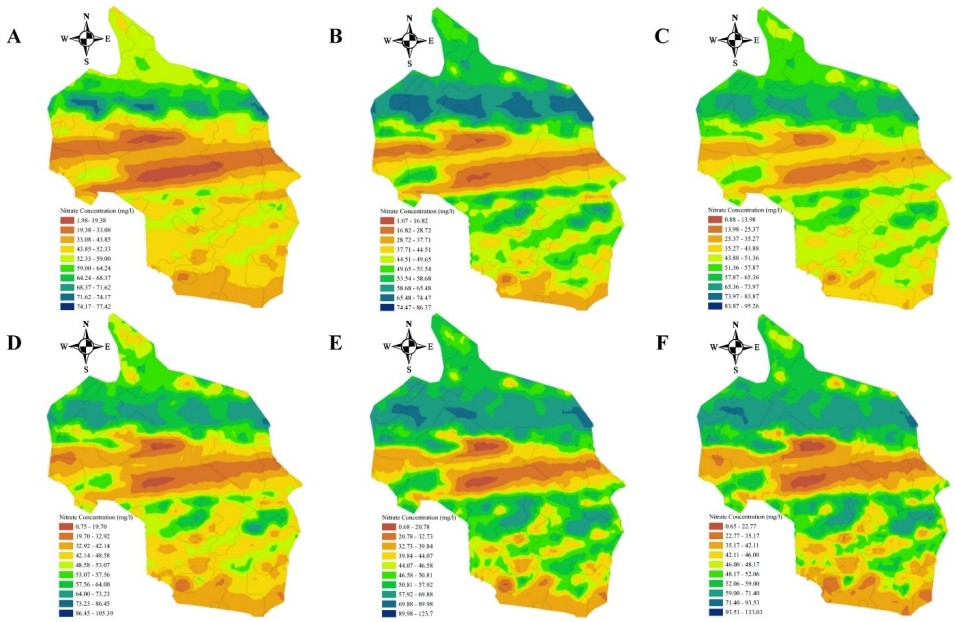

**Figure 4**



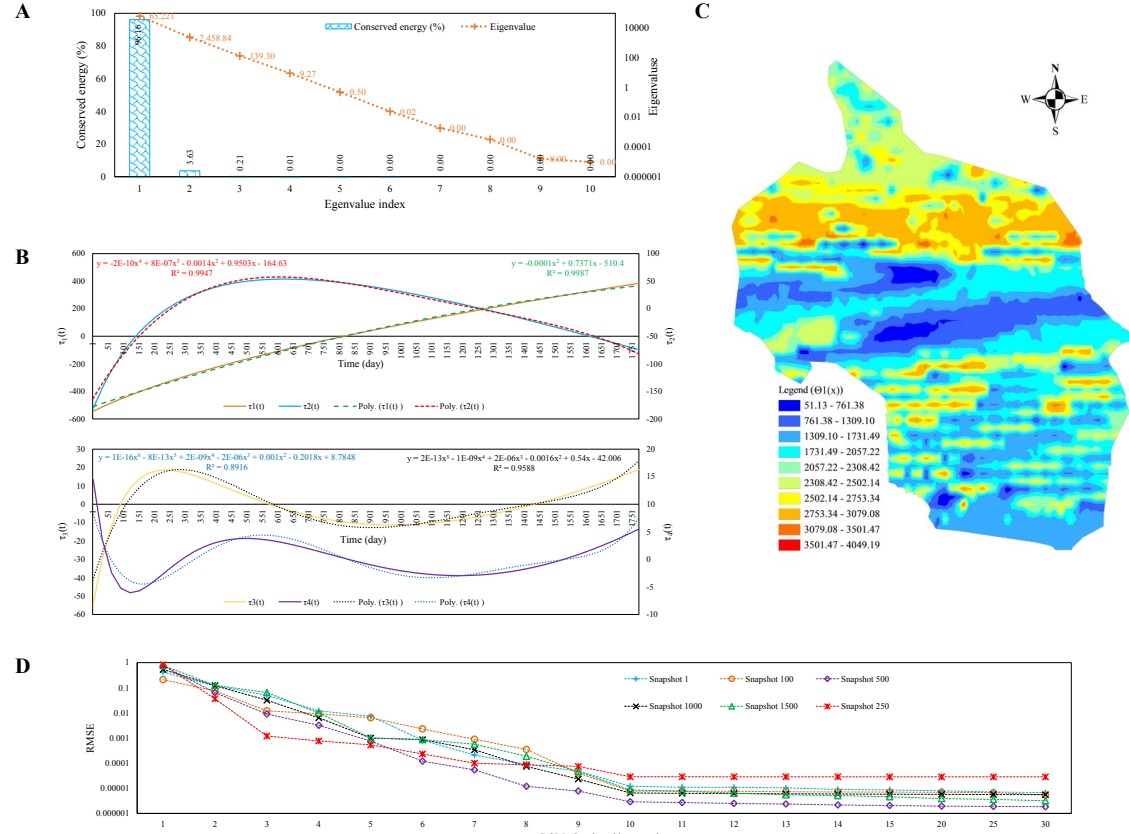

**Figure 5**

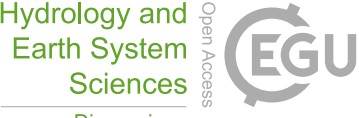



**Figure 6**



**Figure 7**