# Peer review of "A novel model for simulation of nitrate in aquifers"

_Hydrology and Earth System Sciences, 2018_

## Referee Comment (RC1) · Anonymous Referee #1 · 12 Jun 2018

Review of "A novel model for simulation of nitrate in aquifers"

Summary: The manuscript presents a reduced order modeling (ROM) methodology to make predictive model analyses for subsurface solute transport of nitrate. The reduced order model is first developed based on results from a numerical flow and solute transport model simulated using MODFLOW and MT3DMS. The manuscript then compares predictive results generated using the proposed ROM methodology with the predictive results of MT3DMS.

**General Comments**

The main point of the manuscript is the computational efficiency of the proposed ROM methodology. It is claimed to be computationally more efficient than simulating

MT3DMS, however, the manuscript does not provide any comparison of computation time. An appropriate comparison between computing time would include time required for the predictive MT3DMS model, versus time required for ROM analyses that would include calculations to generate eigen values and vectors, matrix computations, and predictive analyses.

It would be beneficial to the reader to also include in the manuscript a discussion about the general applicability and suitability of the ROM methodology, limitations, and the robustness of the ROM predictions. For example, is ROM suitable for predictive scenarios to examine remediation options by adding pumping wells?

It is noteworthy that the ROM methodology is based on a historic numerical model simulated using MT3DMS and therefore, the quality of analyses resulting from ROM is expected to be just as good as the quality of the underlying numerical model, based on which eigen vectors are generated for ROM calculations.

Specific Comments (individual scientific questions/issues)

Below is a list of specific comments that would need to be addressed: - Page 1, line 22: the "simpler structure" of ROM computation is based on matrix calculations but the results are primarily based on MT3DMS computations. I suggest deleting "simpler structure" from the text as that description is misleading.

- Page 1, line 22: provide some numbers to demonstrate "shorter calculation times".

- Page 2, line 9: "information produced by the models is confusing", is an inappropriate statement. It is the modeler's job to understand the meaning of the output that a model generates. Again, the numerical model output is what is used for ROM, which makes it further "confusing", doesn't it?

- Page 2, line 31: "complex mathematical form" and "complex solving methods" for numerical models is presented as a limitation, however, ROM is based on the output from these very "complex" numerical models; ROM, in my opinion adds one more layer

HESSD
of mathematical complexity to the system. If complexity of numerical models is being criticized, ROM stands to be criticized even more. I suggest deleting this line entirely.

- Page 5, line 1: the impact of river recharge is not seen in the head contours; may be the recharge amount is small? I am only stating an observation, this need not be addressed in the manuscript.

- Page 5, line 30: calibration is discussed in detail. My suggestion would be to either shorten the calibration discussion as the focus of the manuscript is the ROM methodology, or include a plot showing the goodness of fit, comparing observed values and simulated results using a scatter plot one for heads and one for concentrations, to complete the calibration discussion.

- Page 6, line 4: incorrect statement, TVD scheme is not a combination of four other methods. I suggest modifying this statement.

- Page 7, line 8-14: the explanation seems unclear. This paragraph is the main feature of this manuscript and needs to be explained better.

- Page 7, line 28: there are several aspects to examine before calling the model well calibrated. It also depends on the objective of the model. In this case, since solute transport is important, getting the gradients and velocities correct becomes important. I am simply pointing this out and the authors may have already examined this aspect but not reported it. This point need not be addressed in the manuscript.

- Page 8, line 5: looking at only the difference can be misleading. Examining timeseries is also important to assess trends.

- Page 8, line 8: Nitrate distribution seems locally contained? Is it because the movement is slow with respect to the simulation period? Again, just an observation. This point need not be addressed in the manuscript.

- Page 10, line 4: the limitations of numerical models listed in the manuscript are arbitrary, as pointed out in some of my previous comments. The only relevant limitation
of numerical models, in context of this manuscript, could be the computation time, but that analyses is not presented in the manuscript.

- Page 10, line 15: the claim that "ROM was superior than MT3DMS" is incorrect. ROM is based on the results generated by MT3DMS. How would that make ROM superior than MT3DMS in terms of quality of results? I suggest deleting this line from the manuscript.

**Technical Corrections**

Below is list of my technical corrections/suggestions:

- Page 1, line 13: delete "Please", the first word in the Abstract.

- Page 1, line 13: high computational cost is a result of long simulation times, not the other way around.

- Page 1, line 16: replace "presents a solution for the problem in ROMs" with "was".

- Page 1, line 20: insert code or program or simulator before "(MT3DMS)".

- Page 1, line 26: consider rearranging the sentence as: "... and activities have resulted in spreading pollution in the aquifers that result in groundwater quality deterioration."

- Page 1, line 28: nitrate is not "often the main concern", but is one of the common contaminants.

- Page 1, line 31: abstract uses GQM, not GQSM. Search the remainder of manuscript and use a consistent acronym.

- Page 3, line 11: "annual evaporation" or "annual potential evaporation"?
- Page 3, line 28: in equation 1, " $\partial$ " is missing in the denominator in 2 places.
- Page 4, line 22: "distribution of hydraulic", the word 'of' is missing.
- Page 4, line 24: consider replacing "are" with "were".
Interactive

comment

- Page 5, line 11: consider replacing "including" with "the availability of".

- Page 5, line 16: what is the difference between gridded network and mesh dimension? Consider clarifying in the text.

---

## Author Comment (AC1) · 27 Jul 2018

Response to Reviewer' Comments on: https://doi.org/10.5194/hess-2018-222

The authors thank the reviewer for the quick response. The manuscript has been improved substantially based on the constructive comments of the reviewer.

Response to Comments of Reviewer #1

General Comment: The main point of the manuscript is the computational efficiency of the proposed ROM methodology. It is claimed to be computationally more efficient than simulating MT3DMS, however, the manuscript does not provide any comparison of computation time. An appropriate comparison between computing time would include time required for the predictive MT3DMS model, versus time required for ROM

analyses that would include calculations to generate eigenvalues and vectors, matrix computations, and predictive analyses. Response: We agree and have now added a new section in the manuscript that provides a comparison between the running times of the developed ROM and MT3DMS {Please See Supplement File, Track Change Version (Page 10: Lines 19 – 27)}. "Considering the computational costs, the developed nitrate ROM was superior (it ran about ten times faster than MT3DMS). More specifically, the computational time for the predictive MT3DMS model was about 89 seconds. This value reduced to nine seconds when we applied the developed ROM for simulation of nitrate in the aquifer using the same computer. Note that the computational time for running the developed ROM model includes calculations to generate matrix computations, eigenvalues and eigenvectors, and predictive analyses."

General Comment: It would be beneficial to the reader to also include in the manuscript a discussion about the general applicability and suitability of the ROM methodology, limitations, and the robustness of the ROM predictions. For example, is ROM suitable for predictive scenarios to examine remediation options by adding pumping wells? Response: We agree and have added a new section in the manuscript that provides useful information about the general applicability, suitability, limitations, and the robustness of the ROM {Please See Supplement File, Track Change Version (Page 10: Lines 19 – 31; Page 11: Lines 1 and 2)}. "Considering the computational costs, the developed nitrate ROM was superior (it ran about ten times faster than MT3DMS). Also, it is noteworthy that the quality of analyses resulting from ROM was as good as the quality of the underlying numerical model. Thus, it can be concluded that the developed ROM model with a simple structure and considerably less running time than MT3DMS appropriately simulated the nitrate concentration in the Karaj Aquifer. Regarding the ROM applications for the future times, all time dependent factors in the aquifer responses, such as hydro-meteorological, hydrogeological, and well operation strategies in the simulation period is expressed by temporal component $\tau(t)$ in the ROM. Thus, the model is applicable for predictive scenarios to examine the options that already existed during the simulation period of the modelling process. This may limit the application of the developed ROM model for scenarios not experienced during the simulation period (e.g., examine remediation options by adding pumping wells). In other words, the model can appropriately memorize historical processes experienced during the simulation period. Time dependent factors are shown in the temporal component $\tau(t)$ (Fig. 5B) although the spatial component $\Theta(x)$ is not time dependent. Also, it is noteworthy that the ROM is developed based on a historic numerical model simulated using MT3DMS and therefore, (1) the quality of analyses resulting from this model is as good as the quality of the underlying numerical model, (2) one needs to have the results of a historic numerical model to develop ROM model although it can be independently applied for the prediction of nitrate."

General Comment: It is noteworthy that the ROM methodology is based on a historic numerical model simulated using MT3DMS and therefore, the quality of analyses resulting from ROM is expected to be just as good as the quality of the underlying numerical model, based on which eigen vectors are generated for ROM calculations. Response: Thanks for the comment. We have now added a new section in the manuscript that provides useful information about the general applicability, suitability, limitations, and the robustness of the ROM {Please See Supplement File, Track Change Version (Page 10: Lines 24 – 27)}. "Also, it is noteworthy that the ROM is developed based on a historic numerical model simulated using MT3DMS and therefore, (1) the quality of analyses resulting from this model is as good as the quality of the underlying numerical model, (2) one needs to have the results of a historic numerical model to develop ROM model although it can be independently applied for the prediction of nitrate."

Comment: Page 1, line 22: the "simpler structure" of ROM computation is based on matrix calculations but the results are primarily based on MT3DMS computations. I suggest deleting "simpler structure" from the text as that description is misleading. Response: We agree and have deleted it and updated the statement {Please See Supplement File, Track Change Version (Page 1: Line 21)}.

Comment: Page 1, line 22: provide some numbers to demonstrate "shorter calculation

times". Response: We have now provided some numbers about the running times of the models {Please See Supplement File, Track Change Version (Page 1: Line 22)}.

Comment: Page 2, line 9: "information produced by the models is confusing", is an inappropriate statement. It is the modeler's job to understand the meaning of the output that a model generates. Again, the numerical model output is what is used for ROM, which makes it further "confusing", doesn't it? Response: Thanks for the comment. We meant the difficult management of large data produced by these models. However, we have now updated this statement {Please See Supplement File, Track Change Version (Page 2: Lines 6 to 8)}.

Comment: Page 2, line 31: "complex mathematical form" and "complex solving methods" for numerical models is presented as a limitation, however, ROM is based on the output from these very "complex" numerical models; ROM, in my opinion adds one more layer. of mathematical complexity to the system. If complexity of numerical models is being criticized, ROM stands to be criticized even more. I suggest deleting this line entirely. Response: Thanks for the comment. We have now deleted this statement entirely {Please See Supplement File, Track Change Version (Page 2: Lines 30 – 33)}.

Comment: Page 5, line 1: the impact of river recharge is not seen in the head contours; may be the recharge amount is small? I am only stating an observation; this need not be addressed in the manuscript. Response: In arid and semi-arid regions like Iran, most of rivers are not permanent and the Karaj river is no exception. In our groundwater modeling, recharge of river is a small amount compared to the deep groundwater flow. In addition, it is varied during the year although it is considered indirectly in the model.

Comment: Page 5, line 30: calibration is discussed in detail. My suggestion would be to either shorten the calibration discussion as the focus of the manuscript is the ROM methodology, or include a plot showing the goodness of fit, comparing observed values and simulated results using a scatter plot one for heads and one for concentrations, to complete the calibration discussion. Response: Thanks for the comment. We have

now shortened this section {Please See Supplement File, Track Change Version (Page 5: Lines 16 – 31; Page 6: Lines 2 – 6)}.

Comment: Page 6, line 4: incorrect statement, TVD scheme is not a combination of four other methods. I suggest modifying this statement. Response: Thanks for the comment. We have now modified this statement {Please See Supplement File, Track Change Version (Page 6: Lines 2 – 6)}.

Comment: Page 7, line 8-14: the explanation seems unclear. This paragraph is the main feature of this manuscript and needs to be explained better. Response: We have now updated this paragraph. Noted all steps for the introduced methodology are given in Figure 2. However, we did our best to properly describe this section by referring to Figure 2 in the revised manuscript {Please See Supplement File, Track Change Version (Page 7: Lines 8 – 27)}. Note that if the ROM developed by few first selected modes (Eq. (8)) didn't satisfy the defined criteria for evaluating the model performance (such as RMSE and MAE), one need to increase the number of modes so that to reach the defined criteria. This procedure is properly shown in Figure 2. However, Eq. (8) has a simple structure with a few modes. Thus, one needs the few first significant modes developed based on simulated historic nitrate concentrations using MT3DMS to regenerate this pollutant in the aquifer. In other words, the ROM is applicable for regeneration of nitrate on the solution domain of MT3DMS and it is not applicable to make prediction in the future. In this study, we aimed to enhance the ROM to make predictions into the future, for time steps with no nitrate simulations from the MT3DMS model. This step is shown in subsection "Model Verification" of Fig. 2. For this purpose, we developed a methodology that enables the ROM to independently simulate nitrate concentrations. According to this, it is necessary to calculate both the spatiotemporal components $\Theta(x)$ and $\tau(t)$ for the future time interval $(t + n)$. Since the component $\Theta$ is a function of space, it does not change in time. Therefore, it is necessary to calculate $\tau$ for future time steps. For this purpose, a regression equation was used to estimate the time variation of this component $(\tau(t+n))$. Having $\tau(t+n)$, the nitrate concentration

for future time steps t+n can be calculated by:

$$\Omega(x t+n)\sum\tau i(t+n)\Theta i(x)  I  N \quad (9)$$

Eq. (9) simulates the nitrate concentration in different parts of the aquifer for future time t+n using the first few significant modes. According to Fig. 2, if the ROM developed didn't satisfy the defined criteria for evaluating the model performance, one need to increase the number of modes by estimating the trend of more numbers of the time dependent components so that the developed ROM satisfies the defined criteria.

Comment: Page 7, line 28: there are several aspects to examine before calling the model well calibrated. It also depends on the objective of the model. In this case, since solute transport is important, getting the gradients and velocities correct becomes important. I am simply pointing this out and the authors may have already examined this aspect but not reported it. This point need not be addressed in the manuscript. Response: Thanks for the comment. Due to main focus of this submission is solute transport, we have ignored to provide more discussion on the model calibration.

Comment: Page 8, line 5: looking at only the difference can be misleading. Examining timeseries is also important to assess trends. Response: We agree and have now added the trends of measured nitrate concentrations and simulated ones by MT3DMS model. We have also added another figure that properly shows the trends of RMSE and MAE averaged over the study area during the calibration period {Please See Supplement File, Track Change Version (Page 8: Lines 17 – 22; Figure 3D to 3G)}.

Comment: Page 8, line 8: Nitrate distribution seems locally contained? Is it because the movement is slow with respect to the simulation period? Again, just an observation. This point need not be addressed in the manuscript. Response: Thanks for the comment. Yes.

Comment: Page 10, line 4: the limitations of numerical models listed in the manuscript are arbitrary, as pointed out in some of my previous comments. The only relevant

limitation of numerical models, in context of this manuscript, could be the computation time, but that analyses is not presented in the manuscript. Response: Thanks for the comment. We have now added a new section in the manuscript that provides useful information about the running times of the models, i.e. ROM and MT3DMS {Please See Supplement File, Track Change Version (Page 10: Lines 19 – 27)}. "Considering the computational costs, the developed nitrate ROM was superior where it was run about ten times faster than MT3DMS. More specifically, the computational time for the predictive MT3DMS model was about 89 seconds. This value reduced to nine seconds when we applied the developed ROM for simulation of nitrate in the aquifer using the same computer. Note that the computational time for running the developed ROM model includes calculations to generate matrix computations, eigenvalues and eigenvectors, and predictive analyses."

Comment: Page 10, line 15: the claim that "ROM was superior than MT3DMS" is incorrect. ROM is based on the results generated by MT3DMS. How would that make ROM superior than MT3DMS in terms of quality of results? I suggest deleting this line from the manuscript. Response: Thanks for the comment. We have now deleted the mentioned line {Please See Supplement File, Track Change Version (Page 11: Lines 13 and 14)}.

Comment: Page 1, line 13: delete "Please", the first word in the Abstract. Response: Thanks for the comment. Agreed and amended {Please See Supplement File, Track Change Version (Page 1: Line 13)}.

Comment: Page 1, line 13: high computational cost is a result of long simulation times, not the other way around. Response: Thanks for the comment. We have now updated this statement {Please See Supplement File, Track Change Version (Page 1: Line 14)}.

Comment: Page 1, line 16: replace "presents a solution for the problem in ROMs" with "was. Response: Thanks for the comment. We have now updated this statement {Please See Supplement File, Track Change Version (Page 1: Line 16)}.
Comment: Page 1, line 20: insert code or program or simulator before "(MT3DMS)". Response: Thanks for the comment. We have now updated this statement {Please See Supplement File, Track Change Version (Page 1: Line 20)}.

Comment: Page 1, line 26: consider rearranging the sentence as: ". . . and activities have resulted in spreading pollution in the aquifers that result in groundwater quality deterioration.". Response: Thanks for the comment. We have now updated this statement {Please See Supplement File, Track Change Version (Page 1: Lines 27 and 28)}.

Comment: Page 1, line 28: nitrate is not "often the main concern", but is one of the common contaminants. Response: Thanks for the comment. We have now updated this statement {Please See Supplement File, Track Change Version (Page 1: Line 29)}.

Comment: Page 1, line 31: abstract uses GQM, not GQSM. Search the remainder of manuscript and use a consistent acronym. Response: Thanks for the comment. We have now check all text of manuscript and corrected this acronym.

Comment: Page 3, line 11: "annual evaporation" or "annual potential evaporation"? Response: Thanks for the comment. We have now updated this statement {Please See Supplement File, Track Change Version (Page 3: Line 10)}.

Comment: Page 3, line 28: in equation 1, "$\partial$" is missing in the denominator in 2 places. Response: Thanks for the comment. We have now corrected this equation {Please See Supplement File, Track Change Version (Page 3: Line 28)}.

Comment: Page 4, line 22: "distribution of hydraulic", the word 'of' is missing. Response: Thanks for the comment. We have now updated this statement {Please See Supplement File, Track Change Version (Page 4: Line 22)}.

Comment: Page 4, line 24: consider replacing "are" with "were". Response: Thanks for the comment. We have now updated this statement {Please See Supplement File, Track Change Version (Page 4: Line 24)}.
Comment: Page 5, line 11: consider replacing "including" with "the availability of". Response: Thanks for the comment. We have now updated this statement {Please See Supplement File, Track Change Version (Page 5: Line 11)}.

Comment: Page 5, line 16: what is the difference between gridded network and mesh dimension? Consider clarifying in the text. Response: Thanks for the comment. It was a typos error. We have now updated this statement {Please See Supplement File, Track Change Version (Page 5: Lines 16 and 17)}.

Please also note the supplement to this comment:
https://www.hydrol-earth-syst-sci-discuss.net/hess-2018-222/hess-2018-222-AC1-supplement.pdf
* * *
[Figure]

Figure 3

17

**Fig. 1.** Figure 3: Difference between simulated and observed groundwater levels during the calibration of MODFLOW: (A) under unsteady state condition for Ghezel-hesar piezometer located in the northwestern par

---

## Referee Comment (RC2) · Anonymous Referee #2 · 4 Aug 2018

The paper proposes a reduced-order modeling approach for the simulation of nitrate concentration in groundwater. The proposed ROM should reduce the computational burden associated with the solution of flow and transport equations, while providing accurate results of concentration. Results obtained using a full system model (MODFLOW+MT3DMS) to simulate the nitrate concentration in a relatively small aquifer in Iran (Karaj Aquifer) are compared against ROM results to prove the accuracy of the solution.

I have two main concerns that strongly reduce the impact of the proposed paper.

- The aquifer considered is an unconfined aquifer. However, the model used for the simulations, MODFLOW, is a model for confined aquifers (see eq. 1). The

unsaturated zone above the water table may strongly impact the transport of nitrates from the surface to the groundwater. Moreover, the unsaturated zone might be strongly impacted by evapotranspiration. These modelling details should be clearly stated as limitations of the proposed study. Other modelling details are missing in the description of the transport problem, in particular, the handling of the boundary conditions (Dirichlet or Neumann?), and the assigned nitrate concentrations. From Figure 4 (I assume these are modelling results, but it is not specified), the spatial distribution of the nitrate concentration seems to reflect concentration in the recharge, since I cannot see a clear impact of the transport along the flow velocities. Thus, it is fundamental to clearly show (maybe n the SI) the distribution of concentration in input and its temporal variation.

- The Authors are not using POD to assemble a reduced order model, but to build a regression model. One of the peculiarities of projection-based ROMs is that they are obtained from the physical equations governing the system. In fact, the model equations are typically projected in the space of the principal components (or along the main modes, or basis functions). In this way, the ROM can take into account changes in the input variables (e.g., changes in flow velocities or contaminant concentration) and project them in the reduced space. The method used by the Authors, instead, is not projecting the model equations. During the calibration, the Authors are simply analyzing how well the concentrations can be expressed as a linear combination of the modes: this is evident by the fact that their solution is not linked to the physical variables in input. Of course, few modes are sufficient to reproduce the modeled concentrations in this test case, especially because the spatiotemporal variations in concentration are small (see figure 4). During the projection, the coefficients of the modes, $\tau(t)$, are obtained by a regression on the previously computed coefficients. The proposed novel approach in reality is simply building a statistical spatiotemporal model that approximate past results of the physical model. While this approach may work for

short time windows and systems with small temporal variations in concentration, the proposed methodology cannot consider changes in the input variables during the projection, since the model is not directly based on the projection of the model equations. Thus, possible changes in flow velocities or in the input concentrations cannot be included as part of future scenarios. These points are not clear when reading the manuscript.

The methodology presented is simply a statistical extrapolation of future projections from the full-system model solutions. For this reason, I cannot see the novelty of the proposed approach. Title, abstract and Conclusions should be revised accordingly. The conclusions of the paper would be much stronger if the Authors could use a real ROM, based on the projection of the model equations.

In the following, please find other minor considerations. Most figures should be revised: the figure quality is low and the font size used is too small. Finally, I recommend revising the text with a mother-language English speaker.

**Minor comments**

**Abstract**

P1, L15: 'including a simple structure'
what does it means? The goal of ROMs is to pass from high dimensional systems to low systems, possibly linear.

P1, L16: The methodology presented is not 'a solution for the problem', since it does not provide a solution to the transport equation. The proposed model is simply extrapolating future projection from model results. Please revise the sentece.

P1, L18: 'The dominant modes . . . was calculate'
Replace 'was' with 'were'

P1, L17: 'ROMs developed with eigenvectors, to make predictions into the future' Not clear. This sentence should be revised, as the Authors are not using a traditional ROM-based on eigenvectors, but a statistical model.

The abstract should specify that the model used is based on a spatiotemporal regression of model results.

**Introduction**

P2, L22: 'a ROM based eigenvectors'
It should be 'a ROM based on eigenvectors'

P2, L25: GQSM simply does not provide dominant modes, but the application of POD to its output generates the dominant modes. This is a standard procedure, I do not see any 'modeling challenge' here.

P2 L29: There are many applications of POD to transport problems in groundwater. For example, please see:

- Li et al. 2011, 'Numerical simulation based on POD for two-dimensional solute transport problems', Applied Mathematical Modelling

- Rizzo et al. 2018, 'Adaptive POD model reduction for solute transport in heterogeneous porous media', Computational Geoscience.

**2.1 Methods**

P3, L19: It would be interesting to see the data (or to have a reference) showing the degradation of the groundwater quality for the studied area.
P3, L20: The observation wells are shown in Figure S2, not S1

P3, L21: in the description of the study area, it would be important to describe the sources of nitrate for this aquifer and show the available concentration data (maybe in the SI).

P4, L4: Please move the part '$mu_i$ is the seepage or linear pore...' at the end of the sentence, so that it appears just before the associated equation, Eq. 3.

P4, L19: which are the values of the head associated to these Dirichlet boundary conditions? Figure S1 shows that the red boundaries are not along an iso-water table line. Thus, how the imposed head varies along the boundary? As the level table drastically decreased in the past years, are these boundary conditions changing in time? These points should be clarified in the manuscript.

P4, L21: 'For the other boundaries there is no interaction between the adjacent aquifers'. It is more accurate to say that these are 'no flow boundaries conditions'.

P4, L22: 'The spatial distribution hydraulic conductivity'.
Missing 'of the'.

P4, L27: 'The evaporation from groundwater was considered negligible'. This is one of the main limitations in this modelling setup. The Authors are considering an unconfined aquifer, but they are using a model that considers only the saturated part of the aquifer (Eq. 1). Unconfined aquifers can have several meters of variably saturated

soil that strongly interact with precipitation and evapotranspiration and that should be considered for the transport of contaminants.

P5, L5: how was this initial condition determined? Which data on nitrate is available?

P5, L12: 'meteorological, hydrological, geological, hydrogeological, and water quality data'. At which frequency and spatial resolution are these data available? Which are the sources? In particular, which are the hydrological and water quality data used for the calibration and validation of the models?

P5, L28: the flow model equation (eq. 1) used the specific storage $S_s$. However, here the Authors calibrate the specific yield. Please, modify the text in a consiste way.

P5, L29: It is not clear how the Authors pass from the calibrated recharge to the time-varying recharge for the transient simulation.

P6, L5: it is not clear how the spatiotemporal variation of the nitrate in input is considered. Is this a fixed percentage of the input flows? Are concentration in the recharge and concentration in the domain boundaries modeled in the same way? (Dirichlet or Neumann boundary conditions?

P6, Section 2.4 I find the description of the POD and of the ROM model quite difficult to follow and with many missing details. The steps for assembling the ROM using POD are standard: collection of snapshots, computation of the principal components (or modes) by using SVD of snapshot matrix. Then, the model solution (vector of concentrations depending on time) is approximated by a linear combination of the first modes. The coefficients of this linear combination, which depend on time, are obtained by projecting the model equation (eq. 2) in the space of the principal components. In

the procedure described by the Authors, it is not clear which variables represent the snapshots of concentrations? What is, in formulas, matrix $M_i, j$? What are matrices $\Omega$ and $\Omega^{(i)}$? Different indices should be used to indicate snapshots/eigenvectors/modes. Please, clarify the spatial dimension of each vector.

P6, Eq. 4: variable $(x)$ should appear on both sides of the equation. The index $(i)$ should appear also for the eigenvector . Moreover, it would be useful to introduce an index to link the particular mode $\Phi$ to the particular eigenvector .

P6, Eq. 6: What is $M_i, j$ in relation to $\Omega$?

P6, Eq. 7: What is $\Phi_i$? This notation has not been defined.

P7, Eq. 2: Here I am not sure to understand, and I think this is due to the fact that eq. (4) is not clear. The modes are represented by vectors $\Phi$; eigenvectors by vectors . To each eigenvalue in eq. (6) is associated one eigenvector and, thus, one mode by eq. (4). Please clarify the text.

P7, Eq. 8: missing $i = 1$ in the summation

P7, L8-16: in the procedure described up to now, the Authors are not using a real ROM. In fact, they are simply projecting the concentrations from the full model space, to the space of the modes, as it is described in Eq. (7) and (8). Usually ROM models use a projected version of the model equations (Eq. 2), so that the variations in the model input data (recharge, boundary conditions) can be directly considered in the ROM future projections. Here, the real ROM model is provided by a regression model

on the time variation of the coefficient $\tau$ (P7, L13). However, there is no information on how this regression is performed and how the input data (e.g. spatiotemporal variation of recharge) are considered in the ROM.

**Results**

P7, L20: please specify the algorithm was used for calibration of the models.

P7, L22: the two piezometers selected to present the results (Figures 3A, 3B) are really close to the imposed boundary conditions. It would be more interesting to see results for an internal piezometer (e.g., Shah Abassi or Chaman).

P7, L23: change 'observations well' in 'observation wells'.

P8, L9: Figures 4A – 4F show also that there is not much transport of nitrates along the flow direction. It seems that, at least for the temporal scale selected, nitrate concentration in the groundwater reflects the concentrations of the recharge. Is this aspect captured by the model?

P8, L16: In methods, $\Omega$ is a function of $x$, not a matrix.

**Conclusions**

P10, L5: Following the considerations done in the introduction, the main limitation of GQSMs is the computational burden. The fact that POD is needed to find the modes of the concentration is not a novelty, nor a limitation of GQSMs.

[Figure]

P10, L7-10: the authors are not developing a ROM here. The computed concentrations are projections of the results of the full system model along the main modes. For this reason they have a small error. But there is not a model behind.

P10, L15-16: The ROM developed is simply a statistical model based on the regression on the results of the physically based model. This is not superior to MT3DMS. Moreover, it cannot be applied for the simulation of other models input or parameters, without before re-computing the MT3DMS solutions.

**Figures**

Figure 1: this figure does not provide useful information to understand the geomorphology of the study area. I suggest replaing this figure by the current Fig S1, with the addiction of a small map showing the basin location within Iran, in the upper left corner.

Figure 2: please revise this figure. The flow chart is quite complex, with too many details that can be avoided. As first, the groundwater and transport part should go on left, as it constitute the first modelling part. The ROM construction should go on the right. The 'START' should be connected only with the flow and transport part, as this is the first modelling step.

- Flow and transport part: the first 6 blocks are standard steps for using MOD-FLOW/MT2DMS. They can be summarized in one block: 'Preparation of input data for MODFLOW and MT3DMS'. I would remove the alternative calibration for steady-state and transient flow here: they have been used one after the other: before calibration of hydraulic conductivity and recharge using steady state assumption. After calibration of specific yield. The last 3 blocks should be in the part of ROM development, and can be summarized in 'Snapshot selection and

extraction from MT3DMS results'.

- ROM part: the currently first 4 blocks can be summarized in 'Computation of principal modes associated to the snapshots'. The block 'Calculation of temporal and spatial . . .' goes after the selection of the number of modes. 'Developing the ROM for. . .' should be 'Solution of the ROM for. . .'. In the 'no alternative' after 'Satisfying the defined criteria', there should be a block 'Add one mode in the ROM space'. The arrow should connect to the block 'Computation of temporal and spatial terms of ROM . . .'.

- Verification part: what does it means 'development of the ROM?'. The ROM has been developed before, as written in the algorithm: Model Construction. Why do you need to develop it again? And how? This is not explained in the diagram.

Figure 3: together with the temporal variation of the error, it would be interesting to see, for each well, the temporal variation of the data (the Authors can use a different scale on the right of each panel). In this way the readers can better understand is the model is capturing the timing of fluctuations of the data.

Figure 4: the legend for the colorbar is too small. All maps should have the same color scale, so that only one big color bar is needed. Please specify in the caption if these maps are modeling results or interpolation of data-point

**Figures SI**

Figures S3, S4: missing measuring units.

Figure S5: Why the zones used to assign values of the hydraulic conductivity (in figure S3) are different from the zones used for the specific yield?

---

## Author Comment (AC2) · 30 Sep 2018

Response to Reviewer' Comments on: https://doi.org/10.5194/hess-2018-222

The authors thank the reviewer for the quick response. The manuscript has been improved substantially based on the constructive comments of the reviewer. Note all responses to the reviewers' comments are added to the manuscript.

Response to Comments of Reviewer #2

Comment #1: The aquifer considered is an unconfined aquifer. However, the model used for the simulations, MODFLOW, is a model for confined aquifers (see eq. 1). The unsaturated zone above the water table may strongly impact the transport of nitrates from the surface to the groundwater. Moreover, the unsaturated zone might be

strongly impacted by evapotranspiration. These modelling details should be clearly stated as limitations of the proposed study. Other modelling details are missing in the description of the transport problem, in particular, the handling of the boundary conditions (Dirichlet or Neumann?), and the assigned nitrate concentrations. From Figure 4 (I assume these are modelling results, but it is not specified), the spatial distribution of the nitrate concentration seems to reflect concentration in the recharge, since I cannot see a clear impact of the transport along the flow velocities. Thus, it is fundamental to clearly show (maybe n the SI) the distribution of concentration in input and its temporal variation. Response: Thank you for your comment. MODFLOW can be used in unconfined aquifers although it conceptually models an unconfined aquifer as a confined system (Carroll et al., 2009). Comparison between MODFLOW and models which are developed for unconfined aquifers shows that approximating the unconfined system in the MODFLOW as a constant-saturated-thickness system (confined system) performs very well (Faunt et al. 2010). As the first step in the groundwater modelling, MODFLOW users should select the aquifer type. In this paper, Karaj aquifer which is an unconfined aquifer has been modelled and the governing equation of unconfined aquifers, i.e. has been solved. Indeed, a mistake was made in presenting the governing equation (Eq. (1)) and now the equation is revised in the manuscript Also, some of the model's limitations are described in the paper as follows: "MODFLOW conceptually models an unconfined aquifer as a constant-saturated-thickness system (confined system). It finds the saturated zone of the unconfined aquifer by calculating h in each grid. It doesn't take in to account the unsaturated zone above the water table, although this vadose zone may impact the transport of pollutant from the surface to the groundwater. Moreover, the unsaturated zone might be impacted by evapotranspiration which is not considered in MODFLOW. Furthermore, MODFLOW uses Dupuit-Forchheimer assumption that the streamlines are horizontal in each column, which is not acceptable for slope more than 0.1 (Wang et al. 2014). Therefore, it is unable to simulate transient water table around pumping gravity wells very well." In addition, boundary conditions of the transport model are explained in the paper as follows (more details

are explained in the comments 10 and 13): "For the transport model no boundary condition is defined because there is no entrance pollutant in the boundaries. The pollutant load enters the domain just due to human activities which are done above the aquifer area. Note that the initial condition is also defined to the model by using observed nitrate concentrations at the observation wells in April 2006. Besides, as you mentioned, Figure 4 shows the modelling results. More details on this figure are presented in Comments # 25 and 36. Carroll, R. W. H., Pohll, G. M., and Hershey, R. L., 2009. An unconfined groundwater model of the Death Valley Regional Flow System and a comparison to its confined predecessor. Journal of Hydrology, 373 (3–4), 316-328. https://doi.org/10.1016/j.jhydrol.2009.05.006 Faunt, C. C., Provost, A. M., Hill, M. C., and Belcher W. R., 2011. Comment on "An unconfined groundwater model of the Death Valley Regional Flow System and a comparison to its confined predecessor" by R.W.H. Carroll, G.M. Pohll and R.L. Hershey [Journal of Hydrology 373/3–4, pp. 316–328], Journal of Hydrology, 397(3-4), 306-309. https://doi.org/10.1016/j.jhydrol.2010.11.038 Wang, Q., Zhan, H., and Tang, Z., 2014. A new package in MODFLOW to simulate unconfined groundwater flow in sloping aquifers, Ground Water, 52(6), 924-35, https://doi.org/10.1111/gwat.12142

Comment #2: The Authors are not using POD to assemble a reduced order model, but to build a regression model. One of the peculiarities of projection-based ROMs is that they are obtained from the physical equations governing the system. In fact, the model equations are typically projected in the space of the principal components (or along the main modes, or basis functions). In this way, the ROM can take into account changes in the input variables (e.g., changes in flow velocities or contaminant concentration) and project them in the reduced space. The method used by the Authors, instead, is not projecting the model equations. During the calibration, the Authors are simply analyzing how well the concentrations can be expressed as a linear combination of the modes: this is evident by the fact that their solution is not linked to the physical variables in input. Of course, few modes are sufficient to reproduce the modeled concentrations in this test case, especially because the spatiotemporal variations in concentration are small (see figure 4). During the projection, the coefficients of the modes, $\tau(t)$, are obtained by a regression on the previously computed coefficients. The proposed novel approach in reality is simply building a statistical spatiotemporal model that approximate past results of the physical model. While this approach may work for short time windows and systems with small temporal variations in concentration, the proposed methodology cannot consider changes in the input variables during the projection, since the model is not directly based on the projection of the model equations. Thus, possible changes in flow velocities or in the input concentrations cannot be included as part of future scenarios. These points are not clear when reading the manuscript. Response: Thanks for the comment. We have now changed the model's name as "POD model". We agree that the proposed model is a statistical spatiotemporal model. But, this is not just a model for approximation of the past results of the physical model. Regarding the developed model applications for the future times (without before re-computing the MT3DMS solutions), all time dependent factors in the aquifer responses, such as hydro-meteorological and hydrogeological variables in the simulation period are expressed by temporal component $\tau(t)$ in the developed predictive POD model. Thus, the model is applicable for future prediction of nitrate in the aquifer under conditions that already existed during the simulation period of the modelling process. In other words, the predictive POD model can appropriately memorize historical processes experienced during the simulation period so that the model captures the dominant modes of nitrate variation in the aquifer. These modes that include space and time dependent terms of nitrate ($\Theta(x)$ and $\tau(t)$, respectively), clearly represent the spatiotemporal variation (STV) of this pollutant in the aquifer. Thus, the changes in the input concentrations are expressed in the time dependent terms of the developed model extrapolated in the future (i.e. $\tau(t+n)$). By considering these, we used the predictive POD model developed by the first four modes that represent more than 99.99% of system energy to predict nitrate in the aquifer. This means that the developed model conserves more than 99.99% of STV of nitrate in the aquifer. Therefore, it can be concluded that the results of developed model for prediction of nitrate match

well the nitrate simulated by MT3DMS. To clearly show this fact, one-year data from April 2012 to March 2013 was employed to compare the developed predictive POD model with the MT3DMS. As described in the manuscript (Section 3.4.2), the absolute error between the developed model and the MT3DMS was less than 0.5 mg/l in the most parts of the aquifer. This clearly shows that the developed model behaves in a similar accuracy as MT3DMS. However, while the developed model is applicable for prediction of nitrate in the future, it cannot be used for evaluation of different scenarios. In fact, inputs used for development of the predictive POD model are $\Theta(x)$ and $\tau(t)$. Although all time dependent factors (such as hydro-meteorological and hydrogeological variables) in the aquifer responses are expressed by $\tau(t)$ in the developed predictive POD model, $\tau(t)$ doesn't individually include these time dependent factors. Therefore, it is not possible to change these time dependent factors in the predictive POD model individually although they are totally expressed by $\tau(t)$. In addition, it is noteworthy that the predictive POD model is developed based on a historic numerical model simulated using MT3DMS and therefore, (1) the quality of analyses resulting from this model is as good as the quality of the underlying numerical model, (2) one needs to have the results of a historic numerical model to develop the predictive POD model although it can be independently applied for the prediction of nitrate." We have now added a section in the manuscript that clearly describes these limitations of the developed POD model."

Comment #3: The methodology presented is simply a statistical extrapolation of future projections from the full-system model solutions. For this reason, I cannot see the novelty of the proposed approach. Title, abstract and Conclusions should be revised accordingly. The conclusions of the paper would be much stronger if the Authors could use a real ROM, based on the projection of the model equations. Response: Thanks for the comment. We have now changed the model's name as "POD model". In this regard, title, abstract and conclusions sections were revised accordingly. A look at the literature demonstrates that some investigators have sought to apply POD to numerical models' output data (or field data) in an effort to develop an appropriate model for investigation of their objective parameters. However, this type of efforts cannot be used to forecast a phenomenon in the future. In fact, the POD method is only capable of regenerating results simulated by numerical models or information produced within a field study's period of investigation (Ashrsfi 2012; Noori et al. 2015; Kheirabadi et al. 2018; Noori et al. 2018). In light of this fact, the main objective of the present study was to develop a POD model which could serve as a tool to forecast nitrate concentration in an important aquifer in Iran, i.e. Karaj Aquifer (without before re-computing the MT3DMS solutions). Therefore, as you mentioned, we just added a linear regression to predict a basis coefficient function $\tau(t)$. We agree with your comment, but no methodological approach has previously applied this idea to forecasting. In other words, although this is a straightforward approach, application of this idea to remedy the limitation of POD for forecasting is new. Ashrafi, K. 2012. Determining of spatial distribution patterns and temporal trends of an air pollutant using proper orthogonal decomposition basis functions, Atmos. Environ., 47, 468-476, https://doi.org/10.1016/j.atmosenv.2011.10.016 Noori, R., Asadi, N., and Deng, Z.: A simple model for simulation of reservoir stratification. J. Hydraul. Res. https://doi.org/10.1080/00221686.2018.1499052 Kheirabadi, H., Noori, R., Samani, J., Adamowski, J.F., Ranjbar, M.H. and Zaker, N.H., 2018. A reduced-order model for the regeneration of surface currents in Gorgan Bay, Iran. Journal of Hydroinformatics. https://doi.org/10.2166/hydro.2018.149

Comment #4: Most figures should be revised: the figure quality is low and the font size used is too small. Finally, I recommend revising the text with a mother-language English speaker. Response: Thanks for the comment. We have now modified figures based on your comments. Also, I asked my colleagues Prof. Ronny Berndetsson from Lund University and Dr. Jan Adamowski from McGill University to modify the text of manuscript.

Comment #5: P1, L15: 'including a simple structure' what does it means? The goal of ROMs is to pass from high dimensional systems to low systems, possibly linear. Response: Thanks for the comment. We mean the simple structure of the developed

predictive POD model that just include both space and time dependent terms ($\Theta(x)$ and $\tau(t)$, respectively) as shown in Eq. (9). However, we have now removed this phrase based on the first reviewer's comments.

Comment #6: P1, L16: The methodology presented is not 'a solution for the problem', since it does not provide a solution to the transport equation. The proposed model is simply extrapolating future projection from model results. Please revise the sentence. Response: Thanks for the comment. We have now revised this statement.

Comment #7: P1, L17: 'ROMs developed with eigenvectors, to make predictions into the future' Not clear. This sentence should be revised, as the Authors are not using a traditional ROM-based on eigenvectors, but a statistical model. Response: Thanks for the comment. We have now revised this statement.

Comment #8: P1, L18: 'The dominant modes : : : was calculate'. Replace 'was' with 'were' Response: Thanks for the comment. We have now revised this statement.

Comment #9: The abstract should specify that the model used is based on a spatiotemporal regression of model results. Response: Thanks for the comment. We have now revised the Abstract section and used POD model instead of ROM.

Comment #10: P2, L22: 'a ROM based eigenvectors' It should be 'a ROM based on eigenvectors' Response: Thanks for the comment. We have now revised this statement.

Comment #11: P2, L25: GQSM simply does not provide dominant modes, but the application of POD to its output generates the dominant modes. This is a standard procedure; I do not see any 'modeling challenge' here. Response: Thanks for the comment. We have now revised this statement.

Comment #12: P2 L29: There are many applications of POD to transport problems in groundwater. For example, please see: • Li et al. 2011, 'Numerical simulation based on POD for two-dimensional solute transport problems', Applied Mathematical

Modelling • Rizzo et al. 2018, 'Adaptive POD model reduction for solute transport in heterogeneous porous media', Computational Geoscience. Response: Thanks for the comment. We have now go through the useful papers and applied them in the manuscript.

Comment #13: P3, L19: It would be interesting to see the data (or to have a reference) showing the degradation of the groundwater quality for the studied area. Response: Thanks for the comment. The following studies are referred to show the degradation of aquifer quality in Section 2.1. "Fazel Tavassol and Gopalakrishna (2014) studied hydrogeochemical parameters to develop water quality index in Karaj aquifer. They concluded that groundwater quality has been impaired by man-made activities, and proper management plan is necessary to protect groundwater resources. Nitrate is one of the main pollutants and a serious problem in Karaj aquifer that penetrates into the groundwater from various sources such as chemical fertilizers, pesticides, and domestic and industrial sewage. Unfortunately, the nitrate pollution zone in this aquifer, with concentrations far beyond the permitted limit (50 mg/L), expands fast. The entire aquifer area has been drinkable in 2000, but with the increase in nitrate concentration, the area with undrinkable water has expanded to 21% in 2003, 24% in 2005, 33% in 2007, 39% in 2009, 43% in 2011 and 44% in 2013 (Chitsazan et al. 2017)." Fazel Tavassol, S., and Gopalakrishna, G. S., : Assesment Of Groundwater Quality Analisis Inkaraj Plain, Albourz Province, Iran , International Journal on Advances in Life Sciences, 4(11), 9089-9096 Chitsazan, M., Mohammad Rezapour Tabari, M., and Eilbeigi, M. 2017. Analysis of temporal and spatial variations in groundwater nitrate and development of its pollution plume: a case study in Karaj aquifer, Environmental Earth Sciences, 76(11). https://doi.org/10.1007/s12665-017-6677-7

Comment #14: P3, L20: The observation wells are shown in Figure S2, not S1 Response: Thanks for the comment. We have now revised it in the manuscript.

Comment #15: P3, L21: in the description of the study area, it would be important to describe the sources of nitrate for this aquifer and show the available concentration

data (maybe in the SI). Response: Thanks for the comment. The main sources of nitrate in the aquifer are agricultural activities, widespread use of latrine wells, and industrial effluents. However, we have now provided a full description on the possible sources of nitrate in the aquifer. According to the approvals of the Iranian Supreme National Security Council, the water quality data of potential drinking water bodies (such as groundwater) is secret. Therefore, publishing the raw data might result in problems for the us.

Comment #16: P4, L4: Please move the part 'mui is the seepage or linear pore: : :' at the end of the sentence, so that it appears just before the associated equation, Eq. 3. Response: Thanks for the comment. Agreed and amended.

Comment #17: P4, L19: which are the values of the head associated to these Dirichlet boundary conditions? Figure S1 shows that the red boundaries are not along an iso-water table line. Thus, how the imposed head varies along the boundary? As the level table drastically decreased in the past years, are these boundary conditions changing in time? These points should be clarified in the manuscript. Response: The head values of the grids on the boundaries are obtained by the Kriging spatial analysis. They are fixed and varied in the steady and unsteady model, respectively.

Comment #18: P4, L21: 'For the other boundaries there is no interaction between the adjacent aquifers'. It is more accurate to say that these are 'no flow boundaries conditions'. Response: Thanks for the comment. We have now modified this statement.

Comment #19: P4, L22: 'The spatial distribution hydraulic conductivity'. Missing 'of the'. Response: Thanks for the comment. We have now revised this statement.

Comment #20: P4, L27: 'The evaporation from groundwater was considered negligible'. This is one of the main limitations in this modelling setup. The Authors are considering an unconfined aquifer, but they are using a model that considers only the saturated part of the aquifer (Eq. 1). Unconfined aquifers can have several meters of variably saturated soil that strongly interact with precipitation and evapotranspiration

and that should be considered for the transport of contaminants Response: Thanks for the comment. MODFLOW calculates the hydraulic head in any grids (columns) and base on the distance between water table and ground surface calculates the amount of evaporation. As noted in the paper, it has been recommended that for deep groundwater (more than 3m depth), evaporation is negligible. Therefore, in this study the evaporation has not been considered, although the recharge due to precipitation has been considered. Noted that, depth to the water table varies from less than 5 m to more than 200 m in the plain. Note that MODFLOW does not consider the unsaturated zone above water table; therefore, evaporation from this zone that affects nitrate transport is neglected. This is one of the model limitations which is noted in the modified paper.

Comment #21: P5, L5: how was this initial condition determined? Which data on nitrate is available? Response: We had data of nitrate concentrations in the observation wells twice a year; although the model has been run monthly. The calibration period of transport model starts in April 2006 (onset of unsteady modelling). By Kriging spatial analysis the concentrations are estimated in each grid and put in the transport model as the initial nitrate concentrations. More details are described in Section 2.3.2.

Comment #22: P5, L12: 'meteorological, hydrological, geological, hydrogeological, and water quality data'. At which frequency and spatial resolution are these data available? Which are the sources? In particular, which are the hydrological and water quality data used for the calibration and validation of the models? Response: Thanks for the comment. This period was selected due to including the most accurate and complete data such as monthly meteorological, hydrological, geological, and hydrogeological data and water quality data that is available twice a year, as well. These data are collected in the observation wells illustrated in Figure S2. Noted that the calibration parameters for the flow model are hydraulic conductivity and specific yield to tune the hydraulic heads in the observation wells and dispersion coefficient for the transport model. In addition, we have cite the source of data in reference list of manuscript.

Comment #23: P5, L28: the flow model equation (eq. 1) used the specific storage

Ss. However, here the Authors calibrate the specific yield. Please, modify the text in a consist way. Response: Thanks for the comment. Indeed, we had a mistake in equation (1). The current equation (1) has been changed to that of an unconfined aquifer in which Sy is the specific yield.

Comment #24: P5, L29: It is not clear how the Authors pass from the calibrated recharge to the time varying recharge for the transient simulation. Response: Thank you for your cautious. It was a typos mistake. The P5 L29 is corrected as "In this process, monthly time series of recharge, initial head and exploitation well data were used as the model input".

Comment #25: P6, L5: it is not clear how the spatiotemporal variation of the nitrate in input is considered. Is this a fixed percentage of the input flows? Are concentration in the recharge and concentration in the domain boundaries modeled in the same way? (Dirichlet or Neumann boundary conditions? Response: The initial concentrations in any grid is introduce to the model in mg/l (the collected data are alike) and based on the flow amount in the recharge area, the injected nitrate load can be calculated. Also, in the transport model there is no entrance nitrate load through boundaries.

Comment #26: P6, Section 2.4 I find the description of the POD and of the ROM model quite difficult to follow and with many missing details. The steps for assembling the ROM using POD are standard: collection of snapshots, computation of the principal components (or modes) by using SVD of snapshot matrix. Then, the model solution (vector of concentrations depending on time) is approximated by a linear combination of the first modes. The coefficients of this linear combination, which depend on time, are obtained by projecting the model equation (eq. 2) in the space of the principal components. In the procedure described by the Authors, it is not clear which variables represent the snapshots of concentrations? What is, in formulas, matrix Mi,j? What are matrices $\Omega$ and $\Omega$(i)? Different indices should be used to indicate snapshots/eigenvectors/modes. Please, clarify the spatial dimension of each vector. Response: Thanks for the comment. We have now updated this section. In this regard,

variable $\Omega$ represents the snapshots of nitrate concentration. Matrix M is a nonnegative Hermitian matrix (we have now added an equation for calculation of this matrix in the manuscript). There is no difference between matrices $\Omega$ and $\Omega$(i) (we have now update this equation). According to Section 2.4. snapshots and eigenvectors were defined by $\Omega$ and ïAÿ, respectively. Also, the both space and time dependent terms of the modes were defined by $\Theta$ and ïAt', respectively. In addition, we have now described the spatial dimension of each vector defined in in Section 2.4.

Comment #27: P6, Eq. 4: variable (x) should appear on both sides of the equation. The index (i) should appear also for the eigenvector. Moreover, it would be useful to introduce an index to link the particular mode $\Phi$ to the particular eigenvector. Response: Thanks for the comment. We have now modified Eq. (4). Variable $\Theta$ is not a mode. It is a space dependent term of mode.

Comment #28: P6, Eq. 6: What is Mi,j in relation to $\Omega$? Response: Thanks for the comment. Matrix M is a nonnegative Hermitian matrix. We have now added an equation for calculation of this matrix in the manuscript that clearly shows the relationship between M and $\Omega$. Based on the new added equation, Matrix M is a function of $\Omega$.

Comment #29: P6, Eq. 7: What is $\Phi$i? This notation has not been defined. Response: Thanks for the comment. This notation is a counter for the number of modes that are equal to snapshots (here, 1772 modes). Therefore, this notation is varied between 1 and 1772.

Comment #30: P7, Eq. 2: Here I am not sure to understand, and I think this is due to the fact that eq. (4) is not clear. The modes are represented by vectors $\Phi$; eigenvectors by vectors. To each eigenvalue in eq. (6) is associated one eigenvector and, thus, one mode by eq. (4). Please clarify the text. Response: Thanks for the comment. We have now revised Section 2.4. All modes are presented by both space and dependent terms $\Theta$ and ïAt', respectively. As you mentioned, each eigenvalue in Eq. (6) is associated with one mode. However, Eq. (4) just present a formula for the space dependent term

Θ of the modes.

Comment #31: P7, Eq. 8: missing i = 1 in the summation. Response: Thanks for the comment. We have now updated Eq. (8).

Comment #32: P7, L8-16: in the procedure described up to now, the Authors are not using a real ROM. In fact, they are simply projecting the concentrations from the full model space, to the space of the modes, as it is described in Eq. (7) and (8). Usually ROM models use a projected version of the model equations (Eq. 2), so that the variations in the model input data (recharge, boundary conditions) can be directly considered in the ROM future projections. Here, the real ROM model is provided by a regression model on the time variation of the coefficient $\tau$ (P7, L13). However, there is no information on how this regression is performed and how the input data (e.g. spatiotemporal variation of recharge) are considered in the ROM. Response: Thanks for the comment. We fitted a function to $\ddot{\tau}1(t)$ to $\ddot{\tau}4(t)$ so that they properly give us the time dependent terms of the model in the future. Thereafter, by considering no changes of Θ(x) in the future, the predictive POD model was developed by just the first four modes that represent more than 99.99% of STV of nitrate in the aquifer. As described in Comment #2, all time dependent factors in the aquifer responses, such as hydro-meteorological and hydrogeological variables in the simulation period are expressed by temporal component $\tau(t)$ in the developed predictive POD model. These modes that include space and time dependent terms of nitrate (Θ(x) and $\tau(t)$, respectively), clearly represent the STV of this pollutant in the aquifer. Thus, the changes in the input concentrations are expressed in the time dependent terms of the developed model extrapolated in the future (i.e. $\tau1(t+n)$ to $\tau4(t+n)$). By considering these, we used the predictive POD model developed by the first four modes that represent more than 99.99% of system energy to predict nitrate in the aquifer. This means that the developed model conserves more than 99.99% of STV of nitrate in the aquifer. Therefore, it can be concluded that the results of developed model for prediction of nitrate match well the nitrate simulated by MT3DMS. To clearly show this fact, one-year data from

April 2012 to March 2013 was employed to compare the developed predictive POD model with the MT3DMS. As described in the manuscript (Section 3.4.2), the absolute error between the developed model and the MT3DMS was less than 0.5 mg/l in the most parts of the aquifer. This clearly shows that the developed model behaves in a similar accuracy as MT3DMS. However, while the developed model is applicable for prediction of nitrate in the future, it cannot be used for evaluation of different scenarios. In fact, inputs used for development of the predictive POD model are Θ(x) and $\tau(t)$. Although all time dependent factors (such as hydro-meteorological and hydrogeological variables) in the aquifer responses are expressed by $\tau(t)$ in the developed predictive POD model, $\tau(t)$ doesn't individually include these time dependent factors. Therefore, it is not possible to change these time dependent factors in the predictive POD model individually although they are totally expressed by $\tau(t)$.

Comment #33: P7, L20: please specify the algorithm was used for calibration of the models. Response: Thanks for the comment. Figure 2 illustrates the calibration process. It is modified to present the parameter calibrated in each step.

Comment #34: P7, L22: the two piezometers selected to present the results (Figures 3A, 3B) are really close to the imposed boundary conditions. It would be more interesting to see results for an internal piezometer (e.g., Shah Abassi or Chaman). Response: Thanks for the comment. We have now updated Figure 3A and 3B with the results of the internal piezometers, i.e. Shah Abassi and Chaman.

Comment #35: P7, L23: change 'observations well' in 'observation wells'. Response: Thanks for the comment. We have now modified this statement.

Comment #36: P8, L9: Figures 4A – 4F show also that there is not much transport of nitrates along the flow direction. It seems that, at least for the temporal scale selected, nitrate concentration in the groundwater reflects the concentrations of the recharge. Is this aspect captured by the model? Response: At the inflow boundaries we have no injected pollutant load because that boundary is a mountain area without any considerable agricultural and industrial activities and most of the activities are located in the middle of the study area. The mentioned figures are the model's results calibrated by measurement; therefore, these figures reflect the real condition of the basin.

Comment #37: P8, L16: In methods, $\Omega$ is a function of x, not a matrix. Response: Thanks for the comment. Whilst each $\Omega$ is a function of x, the set $\Omega$ establishes a matrix with the dimension 1,772 (number of snapshots) $\times$ 4,691 (number of computational cells). We have now revised this statement in the manuscript.

Comment #38: P10, L5: Following the considerations done in the introduction, the main limitation of GQSMs is the computational burden. The fact that POD is needed to find the modes of the concentration is not a novelty, nor a limitation of GQSMs. Response: Thanks for the comment. By considering your valuable comments, we have now revised the text of manuscript. Also, computation of the modes of nitrate in the aquifer is not the main novelty of the present study. As described in Comment #3, the main objective of the present study was to develop a POD model which could serve as a tool to forecast nitrate concentration in an important aquifer in Iran, i.e. Karaj Aquifer without before re-computing the MT3DMS solutions.

Comment #39: P10, L7-10: the authors are not developing a ROM here. The computed concentrations are projections of the results of the full system model along the main modes. For this reason they have a small error. But there is not a model behind. Response: Thanks for the comment. As described in Comment #2, we developed a predictive POD model by the first four modes that represent more than 99.99% of system energy to predict nitrate in the aquifer. This means that the developed model conserves more than 99.99% of STV of nitrate in the aquifer. Also, to clearly show this fact, one-year data from April 2012 to March 2013 was employed to compare the developed predictive POD model with the MT3DMS. As described in the manuscript (Section 3.4.2), the absolute error between the developed model and the MT3DMS was less than 0.5 mg/l in the most parts of the aquifer.

Comment #40: P10, L15-16: The ROM developed is simply a statistical model based on the regression on the results of the physically based model. This is not superior to MT3DMS. Moreover, it cannot be applied for the simulation of other models input or parameters, without before re-computing the MT3DMS solutions. Response: Thanks for the comment. We have now removed this statement: "Therefore, it can be concluded that the developed ROM was superior than MT3DMS". We agree that the proposed model is a statistical spatiotemporal model. But, this is not just a model for approximation of the past results of the physical model. Regarding the developed model applications for the future times (without before re-computing the MT3DMS solutions), all time dependent factors in the aquifer responses, such as hydro-meteorological and hydrogeological variables in the simulation period are expressed by temporal component $\tau(t)$ in the developed predictive POD model. Thus, the model is applicable for future prediction of nitrate in the aquifer under conditions that already existed during the simulation period of the modelling process. In other words, the predictive POD model can appropriately memorize historical processes experienced during the simulation period so that the model captures the dominant modes of nitrate variation in the aquifer. These modes that include space and time dependent terms of nitrate ($\Theta(x)$ and $\tau(t)$, respectively), clearly represent the STV of this pollutant in the aquifer. Thus, the changes in the input concentrations are expressed in the time dependent terms of the developed model extrapolated in the future (i.e. $\tau(t+n)$). By considering these, we used the predictive POD model developed by the first four modes that represent more than 99.99% of system energy to predict nitrate in the aquifer. This means that the developed model conserves more than 99.99% of STV of nitrate in the aquifer. Therefore, it can be concluded that the results of developed model for prediction of nitrate match well the nitrate simulated by MT3DMS. To clearly show this fact, one-year data from April 2012 to March 2013 was employed to compare the developed predictive POD model with the MT3DMS. As described in the manuscript (Section 3.4.2), the absolute error between the developed model and the MT3DMS was less than 0.5 mg/l in the most parts of the aquifer. This clearly shows that the developed model behaves in a

similar accuracy as MT3DMS. However, while the developed model is applicable for prediction of nitrate in the future, it has some limitation same as other models. For example, it cannot be used for evaluation of different scenarios. In fact, inputs used for development of the predictive POD model are $\Theta(x)$ and $\tau(t)$. Although all time dependent factors (such as hydro-meteorological and hydrogeological variables) in the aquifer responses are expressed by $\tau(t)$ in the developed predictive POD model, $\tau(t)$ doesn't individually include these time dependent factors. Therefore, it is not possible to change these time dependent factors in the predictive POD model individually although they are totally expressed by $\tau(t)$.

Comment #41: Figure 1: this figure does not provide useful information to understand the geomorphology of the study area. I suggest replacing this figure by the current Fig S1, with the addition of a small map showing the basin location within Iran, in the upper left corner. Response: Thanks for the comment. We have now replaced this figure with Figure S1 updated with a small map showing the basin location within Iran in the upper left corner.

Comment #42: Figure 2: please revise this figure. The flow chart is quite complex, with too many details that can be avoided. As first, the groundwater and transport part should go on left, as it constitute the first modelling part. The ROM construction should go on the right. The 'START' should be connected only with the flow and transport part, as this is the first modelling step. Flow and transport part: the first 6 blocks are standard steps for using MODFLOW/MT2DMS. They can be summarized in one block: 'Preparation of input data for MODFLOW and MT3DMS'. I would remove the alternative calibration for steady-state and transient flow here: they have been used one after the other: before calibration of hydraulic conductivity and recharge using steady state assumption. After calibration of specific yield. The last 3 blocks should be in the part of ROM development, and can be summarized in 'Snapshot selection and extraction from MT3DMS results'. ROM part: the currently first 4 blocks can be summarized in 'Computation of principal modes associated to the snapshots'. The

block 'Calculation of temporal and spatial : : :' goes after the selection of the number of modes. 'Developing the ROM for: : :' should be 'Solution of the ROM for: : :'. In the 'no alternative' after 'Satisfying the defined criteria', there should be a block 'Add one mode in the ROM space'. The arrow should connect to the block 'Computation of temporal and spatial terms of ROM : : :'. Verification part: what does it means 'development of the ROM?'. The ROM has been developed before, as written in the algorithm: Model Construction. Why do you need to develop it again? And how? This is not explained in the diagram. Response: Thanks for the comment. We have now updated this figure according to your comments. Based on this figure, the POD model was developed at the first step: Model Construction. The developed model at the first step just regenerates the nitrate in the aquifer. Due to the difficulty of the POD model to make predictions into the future, the published research works only aimed at regeneration of objective parameter during the simulation period by now. Therefore, the most important goal of this study was to develop a POD model based eigenvectors that can predict the future concentration of nitrate in the aquifer (beyond the observation period) as described at the second step of Figure 2: Model Verification. However, we have now changes the first and second steps in this figure to "POD Model Construction" and "POD Predictive Model Development", respectively.

Comment #43: Figure 3: together with the temporal variation of the error, it would be interesting to see, for each well, the temporal variation of the data (the Authors can use a different scale on the right of each panel). In this way the readers can better understand is the model is capturing the timing of fluctuations of the data. Response: Thanks for the comment. We have now replaced the temporal variation of error between measured nitrate concentrations and those simulated by MT3DMS with the temporal variation of the nitrate.

Comment #44: Figure 4: the legend for the color bar is too small. All maps should have the same color scale, so that only one big color bar is needed. Please specify in the caption if these maps are modeling results or interpolation of data-point. Response:

Thanks for the comment. We have now revised Figure 4 based on your comments. Note this figure presents the modeling results.

Comment #45: Figures S3, S4: missing measuring units. Response: Thanks for the comment. We have now added the units in Figures S3 and S4.

Comment #46: Figure S5: Why the zones used to assign values of the hydraulic conductivity (in figure S3) are different from the zones used for the specific yield? Response: The model calibration is started according to the available maps of hydraulic conductivity and specific yield. In the calibration process the zones of hydraulic conductivity and specific yield values is not fixed and the values vary till the best result is achieved. As hydraulic conductivity and specific yield values are calibrated in two separate steps, it is not far from expectation to have different zones otherwise the calibration will not be strict.